# Non-Convex Bilevel Optimization with Time-Varying Objective Functions

**Sen Lin**[*]
Department of CS
University of Houston
slin50@central.uh.edu

**Daouda Sow**
Department of ECE
The Ohio State University
sow.53@osu.edu

**Kaiyi Ji**
Department of CSE
University at Buffalo
kaiyiji@buffalo.edu

**Yingbin Liang**
Department of ECE
The Ohio State University
liang889@osu.edu

**Ness Shroff**
Department of ECE & CSE
The Ohio State University
shroff.11@osu.edu

## Abstract

Bilevel optimization has become a powerful tool in a wide variety of machine learning problems. However, the current nonconvex bilevel optimization considers an offline dataset and static functions, which may not work well in emerging online applications with streaming data and time-varying functions. In this work, we study online bilevel optimization (OBO) where the functions can be time-varying and the agent continuously updates the decisions with online streaming data. To deal with the function variations and the unavailability of the true hypergradients in OBO, we propose a single-loop online bilevel optimizer with window averaging (SOBOW), which updates the outer-level decision based on a window average of the most recent hypergradient estimations stored in the memory. Compared to existing algorithms, SOBOW is computationally efficient and does not need to know previous functions. To handle the unique technical difficulties rooted in single-loop update and function variations for OBO, we develop a novel analytical technique that disentangles the complex couplings between decision variables, and carefully controls the hypergradient estimation error. We show that SOBOW can achieve a sublinear bilevel local regret under mild conditions. Extensive experiments across multiple domains corroborate the effectiveness of SOBOW.

## 1 Introduction

Bilevel optimization has attracted significant recent attention, which in general studies the following problem:

$$\min_{x \in \mathbb{R}^{d_1}} \ f(x, y^*(x)) \quad \text{s.t.} \quad y^*(x) = \arg \min_{y \in \mathbb{R}^{d_2}} g(x, y). \tag{1}$$

Here both the outer-level function $f$ and the inner-level function $g$ are continuously differentiable, and the outer optimization problem is solved subject to the optimality of the inner problem. Due to its capability of capturing hierarchical structures in many machine learning problems, this nested optimization framework has been exploited in a wide variety of applications, e.g., meta-learning [4, 31], hyperparameter optimization [13, 3], reinforcement learning [62, 57] and neural architecture search [38, 8].

However, numerous machine learning problems with hierarchical structures are *online* in nature, e.g., online meta-learning [11] and online hyperparameter optimization [63], where the current bilevel

---

[*]The work was done when Sen Lin was in The Ohio State University

37th Conference on Neural Information Processing Systems (NeurIPS 2023).

optimization framework cannot be directly applied due to the following reasons: (1) *(streaming data)* The nature of online streaming data requires decision making on-the-fly with low regret, whereas the offline framework emphasizes more on the quality of the final solution; (2) *(time-varying functions)* The objective functions in online applications can be time-varying because of non-stationary environments and changing tasks, in contrast to static functions considered in Equation (1); (3) *(limited information)* The online learning is nontrivial when the decision maker only has limited information, e.g., regarding the inner-level function, which can be even more challenging with time-varying functions. For example, in wireless network control [37], the controller has to operate under limited knowledge about the time-varying wireless channels (see Appendix).

To reap the success of bilevel optimization in online applications, there is an urgent need to develop a new *online* bilevel optimization (OBO) framework. Generally speaking, OBO considers the scenario where the data comes in an online manner and the agent continuously updates her outer-level decision based on the estimation of the optimal inner-level decision. Both outer-level and inner-level objective functions can be time-varying to capture the data distribution shifts in many online scenarios. Note that OBO is significantly different from single-level online optimization [22], due to the unavailability of the true outer-level objective function composed by $f$ and $y^*$.

The study of OBO was recently initiated in [59], but much of this new framework still remains under-explored and not well understood. In particular, [59] combines offline bilevel optimization with online optimization and proposes an online alternating time-averaged gradient method. Such an approach suffers from several limitations: 1) Multi-step update is required for inner-level decision variable $y_t$ at each time $t$, which can be problematic when only limited information of the inner-level function $g$ is available. 2) The hypergradient estimation at each time requires the knowledge of previous objective functions in a window, and also evaluates current models on each previous function; such a design can be inefficient and infeasible in online scenarios. In this work, we seek to address these limitations and develop a new OBO algorithm that can work efficiently without the knowledge of previous objective functions.

Table 1: Comparison of OBO algorithms. 'HV' product refers to the Hessian inverse-vector product in hypergradient estimation. OAGD estimates the hypergradient by $\frac{1}{W} \sum_{i=0}^{K-1} \eta^i \hat{\nabla} f_{t-i}(x_t, y_{t+1})$, which requires the evaluation of $f_{t-i}$ on current model $(x_t, y_{t+1})$.

| Algorithm | Single-loop update | Study estimation error of HV product | Do not require previous function info |
|---|---|---|---|
| OAGD [59] | ✗ | ✗ | ✗ |
| SOBOW (this paper) | ✓ | ✓ | ✓ |

The main contributions can be summarized as follows.

*(Efficient algorithm design)* We propose a new single-loop online bilevel optimizer with window averaging (SOBOW), which works in a fully online manner with limited information about the objective functions. In contrast to the OAGD algorithm in [59], SOBOW has the following major differences (as summarized in Table 1): (1) *(single-loop update)* Compared to the multi-step updates of the inner-level decision $y_t$ at each round in OAGD, we only require a one-step update of $y_t$, which is more practical for online applications where only limited information about the inner-level function $g_t$ is available. (2) *(estimation error of Hessian inverse-vector product)* Estimating the hypergradient requires the outer-level Hessian inverse-vector product. [59] assumes that the exact Hessian inverse-vector product can be obtained, which can introduce high computational cost. In contrast, we consider a more practical scenario where there could be an estimation error in the Hessian-inverse vector product calculation in solving the linear system. (3) *(window averaged hypergradient descent)* While a window averaged hypergradient estimation is considered in both OAGD and SOBOW, SOBOW is more realistic and efficient than OAGD. Specifically, OAGD requires the knowledge of the most recent objective functions within a window and evaluates the current model on each previous function at every round, whereas SOBOW only stores the historical hypergradient estimation in the memory without any additional knowledge or evaluations about previous functions.

*(Novel regret analysis)* Based on the previous studies in single-level online non-convex optimization [23, 2], we introduce a new bilevel local regret as the performance measure of OBO algorithms. We show that the proposed SOBOW algorithm can achieve a sublinear bilevel local regret under mild conditions. Compared to offline bilevel optimization and OAGD, new technical challenges need to be addressed here: (1) unlike the multi-step update of inner-level variable $y_t$ in OAGD, the single-step update in SOBOW will lead to an inaccurate estimation of the optimal inner-level

decision $y_t^*(x_t)$ and consequently a large estimation error for the hypergradient; this problem can be addressed in offline bilevel optimization by controlling the gap $\|y_t^*(x_t) - y_{t+1}^*(x_{t+1})\|^2$, which depends only on $\|x_t - x_{t+1}\|^2$ for static inner-level functions. But this technique cannot be applied here due to the time-varying $g_t$; (2) the function variations in OBO can blow up the impact of the hypergradient estimation error if not handled appropriately, whereas offline bilevel optimization does not have this issue due to the static functions therein. Towards this end, we appropriately control the estimation error of the Hessian inverse-vector product at each round, and disentangle the complex couplings between the decision variables through a three-level analysis. This enables the control of the hypergradient estimation error, by using a decaying coefficient to diminish the impact of the large inner-level estimation error and leveraging the historical information to smooth the update of the outer-level decision.

(*Extensive experimental evaluations*) OBO has the potential to be used in various online applications, by capturing the hierarchical structures therein in an online manner. In this work, the experimental results clearly validate the effectiveness of SOBOW in online hyperparameter optimization and online hyper-representation learning.

## 2    Related Work

*Online optimization*  Online optimization has been extensively studied for strongly convex and convex functions in terms of both static regret, e.g., [68, 21, 47] and dynamic regret, e.g., [5, 48, 61, 64, 66]. Recently, there has been increasing interest in studying online optimization for non-convex functions, e.g., [1, 58, 34, 51, 15], where minimizing the standard definitions of regret is computationally intractable. Specifically, [23] introduced a notion of local regret in the spirit of the optimality measure in non-convex optimization, and developed an algorithm that averages the gradients of the most recent loss functions evaluated in the current model. [2] proposed a dynamic local regret to handle the distribution shift and also a computationally efficient SGD update for achieving sublinear regret. [24] and [25] studied the zeroth-order online non-convex optimization where the agent only has access to the actual loss incurred at each round.

*Offline bilevel optimization*  Bilevel optimization was first introduced in the seminal work [6]. Following this work, a number of algorithms have been developed to solve the bilevel optimization problem. Initially, the bilevel problem was reformulated into a single-level constrained problem based on the optimality conditions of the inner-level problem [20, 55, 43, 49], which typically involves many constraints and is difficult to implement in machine learning problems. Recently, gradient-based bilevel optimization algorithms have attracted much attention due to their simplicity and efficiency. These can be roughly classified into two categories, the approximate implicit differentiation (AID) based approach [10, 52, 16, 14, 17, 42, 32] and the iterative differentiation (ITD) based approach [45, 12, 53, 44, 17]. Bilevel optimization has also been studied very recently for the cases with stochastic objective functions [14, 31, 7, 33, 18] and multiple inner minima [35, 40, 39, 56]. Notably, a novel value-function based method was first proposed in [39] to deal with the non-convexity of the inner-level functions. Some recent studies, e.g., [36, 9, 26], have also explored single-level algorithms for offline bilevel optimization with time-invariant objective functions, whereas the time-varying functions in online bilevel optimization make the hypergradient estimation error control more challenging for single-loop updates.

*Online bilevel optimization*  The investigation of OBO is still in the very early stage, and to the best of our knowledge, [59] is the only work so far that has studied OBO.

## 3    Online Bilevel Optimization

Following the same spirit as the online optimization in [22], the decisions are made iteratively in OBO without knowing their outcomes at the time of decision-making. Let $T$ denote the total number of rounds in OBO. Define $x_t \in \mathcal{X} \subset \mathbb{R}^{d_1}$ and $f_t : \mathcal{X} \times \mathbb{R}^{d_2}$ as the decision variable and the online function for the outer level problem, respectively. Define $y_t \in \mathbb{R}^{d_2}$ and $g_t \in \mathcal{X} \times \mathbb{R}^{d_2}$ as the decision variable and the online objective function for the inner level problem, respectively. Given the initial values of $(x_1, y_1)$, the general procedure of OBO is described in Algorithm 1.

Let $y_t^*(x) = \arg\min_{y \in \mathbb{R}^{d_2}} g_t(x, y)$ for any $x$. The OBO framework in Algorithm 1 can be interpreted from two different perspectives: (1) (*single-player*) The player makes the decision on $x_t$ without

---

**Algorithm 1** General procedure of OBO

---

1: Initialize decisions $x_1$ and $y_1$
2: **for** $t = 1, ..., T$ **do**
3:  Get information about functions $f_t$ and $g_t$
4:  Update decision $y_{t+1}$ based on $x_t$ and $g_t$
5:  Update decision $x_{t+1}$ based on $y_{t+1}$ and $f_t$
6: **end for**

---

knowing the optimal inner-level decision $y_t^*(x)$. Note, $y_t$ serves as an estimation of $y_t^*(x)$ from the player's perspective based on her knowledge of function $g_t$; (2) (*two-player*) OBO can also be viewed as a leader ($x_t$) and follower ($y_t$) game, where each player considers an online optimization problem and the leader seeks to play against the optimal decision $y_t^*(x)$ of the follower at each round under limited knowledge of $g_t$.

It is worthwhile noting that OBO is quite different from the single-level online optimization. First, the outer-level objective function with respect to (w.r.t) $x_t$, i.e., $f_t(x_t, y_t^*(x_t))$, is not available to update $x_t$, whereas, in standard single-level online optimization, the true loss is revealed immediately after making decisions. Besides, as a composite function of $f_t(x, y)$ and $y_t^*(x)$, $f_t(x, y_t^*(x))$ is *non-convex* in general w.r.t. the outer-level decision variable $x$. Hence, standard regret definitions in online convex optimization [22] are not directly applicable here.

Motivated by the dynamic local regret defined in online non-convex optimization [2], we consider the following bilevel local regret:

$$BLR_w(T) = \sum\nolimits_{t=1}^{T} \|\nabla F_{t,\eta}(x_t, y_t^*(x_t))\|^2 \tag{2}$$

where

$$F_{t,\eta}(x_t, y_t^*(x_t)) = \frac{1}{W} \sum\nolimits_{i=1}^{K-1} \eta^i f_{t-i}(x_{t-i}, y_{t-i}^*(x_{t-i})),$$

and $W = \sum_{i=0}^{K-1} \eta^i$, $\eta \in (0, 1)$, and $f_t(\cdot, \cdot) = 0$ for $t \le 0$. In contrast, the static regret in [59] evaluates the objective at time slot $i$ using variable updates at different time slot $j$, which does not properly characterize the online learning performance of the model update for time-varying functions (see Appendix for more discussion). Intuitively, the regret in Equation (2) is defined as a sliding average of the hypergradients w.r.t. the decision variables at the corresponding instant for all rounds in OBO. This indeed approximately computes the exponential average of the outer-level function values $f_t(x_t, y_t^*(x_t))$ at the corresponding decision variables over a sliding window [2]. Larger weights will be assigned to more recent updates. The objective here is to design efficient OBO algorithms with sublinear bilevel regret $BLR_w(T)$ in $T$, which implies that the outer-level decision is becoming better and closer to the local optima for the outer-level optimization problem at each round. This gradient-norm based regret shares the same spirit as the first-order optimality criterion [14, 32], which is widely used in offline bilevel optimization to characterize the convergence to the local optima.

## 4 Algorithm Design

It is well known that online gradient descent (OGD) [54] has achieved great successes in single-level online optimization. On the other hand, gradient-based methods (e.g., [16, 14, 42, 40, 29]) have become extremely popular for solving offline bilevel optimization due to their high efficiency. Thus, we also study the online gradient descent based algorithm to solve the OBO problem. As mentioned earlier, the unique challenges of OBO should be carefully handled in the algorithm design, including 1) the inaccessibility of the objective function and accurate hypergradients compared to single-level online optimization and 2) the time-varying functions and limited information compared to offline bilevel optimization. To this end, our algorithm includes two major designs, i.e., efficient hypergradient estimation with limited information and window averaged outer-level decision update.

**Hypergradient estimation** In OBO, the exact hypergradient $\nabla f_t(x_t, y_t^*(x_t))$ w.r.t. $x_t$ can be represented as

$$\nabla f_t(x_t, y_t^*(x_t)) = \nabla_x f_t(x_t, y_t^*(x_t)) - \nabla_x \nabla_y g_t(x_t, y_t^*(x_t)) v_t^* \tag{3}$$

where $v_t^*$ solves the linear system $\nabla_y^2 g_t(x_t, y_t^*(x_t))v = \nabla_y f_t(x_t, y_t^*(x_t))$. The optimal inner-level decision $y_t^*(x_t)$ is generally unavailable in OBO. To estimate the hypergradient $\nabla f_t(x_t, y_t^*(x_t))$,

the AID-based approach [10, 14, 32] for offline bilevel optimization can be leveraged here, which will involve the following steps: (1) given $x_t$, run $N$ steps of gradient descent w.r.t. the inner-level objective function $g_t$ to find a good approximation $y_t^N$ close to $y_t^*(x_t)$; (2) given $y_t^N$, obtain $v_t^Q$ by solving $\nabla_y^2 g_t(x_t, y_t^N)v = \nabla_y f_t(x_t, y_t^N)$ with $Q_t$ steps of conjugate gradient. The estimated hypergradient is constructed as

$$\widehat{\nabla} f_t(x_t, y_t^N) = \nabla_x f_t(x_t, y_t^N) - \nabla_x \nabla_y g_t(x_t, y_t^N)v_t^Q. \tag{4}$$

Nevertheless, the $N$ steps gradient descent for estimating $y_t^*(x_t)$ require multiple inquiries about the inner-level function $g_t$, which can be inefficient and infeasible for online applications. For the algorithm being used in more practical scenarios with limited information about $g_t$, we consider the extreme case where $N = 1$, i.e.,

$$y_{t+1} = y_t^1 = y_t - \alpha \nabla_y g_t(x_t, y_t), \tag{5}$$

where $\alpha$ is the inner-level step size. This would lead to an inaccurate estimation of $y_t^*(x_t)$, which can pose critical challenges for making satisfying outer-level decisions in OBO due to the unreliable hypergradient estimation, especially when the objective functions are time-varying.

**Window averaged decision update** To deal with the non-convex and time-varying functions, inspired by time-smoothed gradient descent in online non-convex optimization [23, 2, 67, 19], we consider a time-smoothed hypergradient descent for updating the outer-level decision variable $x_t$:

$$x_{t+1} = \Pi_{\mathcal{X}}(x_t - \beta \widehat{\nabla} F_{t,\eta}(x_t, y_{t+1})) \tag{6}$$

where $\Pi_{\mathcal{X}}$ is the projection onto the set $\mathcal{X}$, $\beta$ is the outer-level step size and

$$\widehat{\nabla} F_{t,\eta}(x_t, y_{t+1}) = \frac{1}{W} \sum_{i=1}^{K-1} \eta^i \widehat{\nabla} f_{t-i}(x_{t-i}, y_{t+1-i}). \tag{7}$$

Here $\widehat{\nabla} f_{t-i}(x_{t-i}, y_{t+1-i})$ is the hypergradient estimation at the round $t - i$, as in Equation (4). Intuitively, the update of the current $x_t$ is smoothed by the historical hypergradient estimations w.r.t. the decision variables at that time, which is particularly important here for OBO due to the following reasons: (1) To compute the averaged $\widehat{\nabla} F_{t,\eta}(x_t, y_{t+1})$ at each round $t$, we only store the hypergradient estimation for previous rounds in the memory, i.e., store $\widehat{\nabla} f_i(x_i, y_{i+1})$ at each round $i$, and estimate the current hypergradient $\widehat{\nabla} f_t(x_t, y_{t+1})$ using the available information at current round $t$. Compared to OAGD in [59], there is no need to access to the previous outer-level and inner-level objective functions and evaluate the current decisions on those functions, which is clearly more efficient and practical for online applications. (2) Leveraging the historical information in the window to update the current decision is helpful to deal with the inaccurate hypergradient estimation for the current round, especially under mild function variations. This indeed shares the same rationale with using past tasks to facilitate forward knowledge transfer in online meta-learning [11] and continual learning [41], for better decision making in the current task and improving the overall performance in non-stationary environments.

Building upon these two major components, we can have our main OBO algorithm, Single-loop Online Bilevel Optimizer with Window averaging (SOBOW), as summarized in Algorithm 2. At the round $t$, we first estimate $y_{t+1}$ as in Equation (5) given $x_t$ and $y_t$, which will be next leveraged to solve the linear system and construct the hypergradient estimation as in Equation (4). Based on the historical hypergradient estimations stored in the memory for previous rounds, we next update $x_t$ based on Equation (6).

## 5 Theoretical Analysis

In this section, we provide the theoretical analysis of the regret bound for SOBOW.

### 5.1 Technical Assumptions

Let $z = (x, y)$. Before the regret analysis, we first make the following assumptions.

**Assumption 5.1.** The inner-level function $g_t(x, y)$ is $\mu_g$-strongly convex w.r.t. $y$, and the composite objective function $f_t(x, y_t^*(x))$ is non-convex w.r.t $x$.

**Assumption 5.2.** The following conditions hold for objective functions $f_t(z)$ and $g_t(z)$, $\forall t \in [1, T]$: (1) The function $f_t(z)$ is $L_0$-Lipschitz continuous; (2) $\nabla f_t(z)$ and $\nabla g_t(z)$ are $L_1$-Lipschitiz continuous; (3) The high-order derivatives $\nabla_x \nabla_y g_t(z)$ and $\nabla_y^2 g_t(z)$ are $L_2$-Lipschitz continuous.

**Algorithm 2** Single-loop Online Bilevel Optimizer with Window averaging (SOBOW)

---

1: Initialize decisions $x_1, y_1, v^0$
2: **for** $t = 1, ..., T$ **do**
3:     Get information about functions $f_t$ and $g_t$
4:     Update $y_{t+1}$ based on Equation (5)
5:     Solve the linear system $\nabla_y^2 g_t(x_t, y_{t+1})v = \nabla_y f_t(x_t, y_{t+1})$ using $Q_t$ steps of conjugate gradient starting from a fixed point $v^0$ with stepsize $\lambda$ to obtain $v_t^Q$
6:     Construct the hypergradient $\widehat{\nabla} f_t(x_t, y_{t+1})$ based on Equation (4) with $y_t^N = y_{t+1}$
7:     Store $\widehat{\nabla} f_t(x_t, y_{t+1})$ in the memory
8:     Update $x_{t+1}$ based on Equation (6)
9: **end for**

---

Note that both Assumption 5.1 and Assumption 5.2 are standard and widely used in the literature of bilevel optimization, e.g., [52, 14, 29, 59, 27].

**Assumption 5.3.** For any $t \in [1, T]$, the function $f_t(x, y_t^*(x))$ is bounded, i.e., $|f_t(x, y_t^*(x))| \le M$ with $M > 0$. Besides, the closed convex set $\mathcal{X}$ is bounded, i.e., $\|x - x'\| \le D$ with $D > 0$, for any $x$ and $x'$ in $\mathcal{X}$.

Assumption 5.3 on the boundedness of the objection functions is also standard in the literature of non-convex optimization, e.g., [23, 2, 50, 59]. Moreover, to guarantee the boundedness of the hypergradient estimation error, previous studies (e.g., [14, 28, 17]) in offline bilevel optimization usually assume that the gradient norm $\|\nabla f(z)\|$ is bounded from above for all $z$. In this work, we make a weaker assumption on the feasibility of $\nabla_y f_t(x, y_t^*(x))$, which generally holds since $y_t^*(x)$ is usually assumed to be bounded in bilevel optimization:

**Assumption 5.4.** There exists at least one $\hat{x} \in \mathcal{X}$ such that $\|\nabla_y f_t(\hat{x}, y_t^*(\hat{x}))\| \le \rho$ where $\rho > 0$ is some constant.

## 5.2 Theoretical Results

**Technical challenges in analysis:** To analyze the regret performance of SOBOW, several key and unique technical challenges need to be addressed, compared to offline bilevel optimization and OAGD in [59]: (1) unlike the multi-step update of inner-level variable $y_t$ in OAGD, the single-step update in SOBOW will lead to an inaccurate estimation of $y_t^*(x_t)$ and consequently a large estimation error for the hypergradient $\nabla f_t(x_t, y_t^*(x_t))$; this problem can be addressed in offline bilevel optimization by controlling the gap $\|y_t^*(x_t) - y_{t+1}^*(x_{t+1})\|^2$, which depends only on $\|x_t - x_{t+1}\|^2$ for static inner-level functions, but this technique cannot be applied here due to the time-varying $g_t$; (2) the function variations in OBO can blow up the impact of the hypergradient estimation error if not handled appropriately, whereas offline bilevel optimization does not have this issue due to the static functions therein; (3) a new three-level analysis is required to understand the involved couplings among the estimation errors about $v_t$, $y_t$ and $x_t$ in online learning.

Towards this end, the very first step is to understand the estimation error of the optimal solution $v_t^*$ to the linear system $\nabla_y^2 g_t(x_t, y_t^*(x_t))v = \nabla_y f_t(x_t, y_t^*(x_t))$. Here $v_t^* = (\nabla_y^2 g_t(x_t, y_t^*(x_t)))^{-1} \nabla_y f_t(x_t, y_t^*(x_t))$. We have the following lemma about $\|v_t^Q - v_t^*\|^2$, where $v_t^Q$ is obtained by solving $\nabla_y^2 g_t(x_t, y_{t+1})v = \nabla_y f_t(x_t, y_{t+1})$ using $Q_t$ steps of conjugate gradient.

**Lemma 5.5.** *Suppose Assumptions 5.1-5.4 hold, $\lambda \le \frac{1}{L_1}$, $\alpha \le \frac{1}{L_1}$ and $Q_{t+1} - Q_t \ge \frac{\log(1 - \frac{\alpha \mu_g}{2})}{2 \log(1 - \lambda \mu_g)}$. We can have that*

$$\|v_t^Q - v_t^*\|^2 \le c_2 \|y_{t+1} - y_t^*(x_t)\|^2 + \epsilon_t^2$$

*where $c_2 > 0$ is some constant and the error $\epsilon_t^2$ decays with t, i.e., $\epsilon_{t+1}^2 \le (1 - \alpha \mu_g/2)\epsilon_t^2$.*

Lemma 5.5 characterizes the estimation error $\|v_t^Q - v_t^*\|^2$ in a neat way, by constructing an upper bound with the estimation error of the inner-level optimal decision $y_t^*(x_t)$, i.e., $\|y_{t+1} - y_t^*(x_t)\|^2$, and a decaying error term $\epsilon_t^2$. The way of controlling $\|v_t^Q - v_t^*\|^2$ here is particularly important, which not only clarifies the coupling between $v_t$ and $y_t$ but also helps to control the hypergradient estimation

error. Note that solving the linear system with a larger $Q_t$ does not require more information about the inner-level function, and the introduced computation cost can be negligible, because the conjugate gradient only involves Hessian-vector product which can be efficiently computed.

Next we seek to bound the hypergradient estimation error $\|\nabla f_t(x_t, y_t^*(x_t)) - \widehat{\nabla} f_t(x_t, y_{t+1})\|^2$ at the round $t$. Intuitively, the hypergradient estimation error depends on both $\|y_{t+1} - y_t^*(x_t)\|^2$ and $\|v_t^Q - v_t^*\|^2$. Building upon Lemma 5.5, this dependence can be shifted to the joint error of $\|y_{t+1} - y_t^*(x_t)\|^2$ and $\epsilon_t^2$, which contains iteratively decreasing components after careful manipulations. Specifically, let $G_1 = 1 + c_2 + \frac{L_2^2(\rho\mu_g + DL_1^2 + DL_1\mu_g)^2}{L_1^2\mu_g^4}$ and $G_2 = 2G_1(1 + \frac{2}{\alpha\mu_g})(1 - \alpha\mu_g)$. We have the following theorem to characterize the hypergradient estimation error.

**Theorem 5.6.** *Suppose that Assumptions 5.1-5.4 hold, $\lambda \leq \frac{1}{L_1}$, $\alpha \leq \frac{1}{L_1}$ and $Q_t - Q_{t-1} \geq \frac{\log(1 - \frac{\alpha\mu_g}{2})}{2\log(1 - \lambda\mu_g)}$. For $t \in [2, T]$, we can bound the hypergradient estimation error as follows:*

$$\|\nabla f_t(x_t, y_t^*(x_t)) - \widehat{\nabla} f_t(x_t, y_{t+1})\|^2 \leq 3L_1^2 \bigg\{ \frac{2L_1^2 G_2}{\mu_g^2} \sum_{j=0}^{t-2} \left(1 - \frac{\alpha\mu_g}{2}\right)^j \|x_{t-1-j} - x_{t-j}\|^2$$

$$+ G_2 \sum_{j=0}^{t-2} \left(1 - \frac{\alpha\mu_g}{2}\right)^j \|y_{t-1-j}^*(x_{t-1-j}) - y_{t-j}^*(x_{t-1-j})\|^2 + \left(1 - \frac{\alpha\mu_g}{2}\right)^{t-1} \delta_1 \bigg\}$$

*where $\delta_1 = G_1 \|y_2 - y_1^*(x_1)\|^2 + \epsilon_1^2$.*

As shown in Theorem 5.6, the upper bound of the hypergradient estimation error includes three terms: (1) The first term decays with $t$, which captures the iteratively decreasing component in the joint error of $\|y_{t+1} - y_t^*(x_t)\|^2$ and $\epsilon_t^2$; (2) The second term characterizes the dependence on the variation of the outer-level decision between adjacent rounds in the history; (3) The third term characterizes the dependence on the variation of the optimal inner-level decision between adjacent rounds.

To control the hypergradient estimation error as in Theorem 5.6, the key idea is to decouple the source of the estimation error $\|y_{t+1} - y_t^*(x_t)\|^2$ at the current round into three different components, i.e., $\|y_t - y_{t-1}^*(x_{t-1})\|^2$ for the previous round, the variation of the out-level decision, and the variation of the optimal inner-level decision. Since the inner-level estimation error is large due to the single-step update, we diminish its impact through a decaying coefficient, which inevitably enlarges the impact of the other two components. The variation of the optimal inner-level decision is due to nature of the OBO problem, which cannot be controlled. One has to impose some regularity constraints on this variation to achieve a sublinear regret, in the same spirit to the regularities on functional variations widely used in the dynamic regret literature (e.g., [5, 66]). Therefore, the key point now becomes the control of the variation of the out-level decision $\|x_{t-1-j} - x_{t-j}\|^2$, which can be achieved through the window averaged update of $x_t$ in Equation (6). Intuitively, by leveraging the historical information, the window averaged hypergradient in Equation (7) smooths the outer-level decision update, which serves as a better update direction compared to the deviated single-round estimation $\widehat{\nabla} f_t(x_t, y_{t+1})$. Before presenting the main result, we first introduce the following definitions to characterize the variations of the objective function $f_t(\cdot, y_t^*(\cdot))$ and the optimal inner-level decision $y_t^*(\cdot)$ in OBO, respectively:

$$V_{1,T} = \sum_{t=1}^{T} \sup_x [f_{t+1}(x, y_{t+1}^*(x)) - f_t(x, y_t^*(x))], \quad H_{2,T} = \sum_{t=2}^{T} \sup_x \|y_{t-1}^*(x) - y_t^*(x)\|^2.$$

Intuitively, $V_{1,T}$ measures the overall fluctuations between the adjacent objective functions in all rounds under the same outer-level decision variable, and $H_{2,T}$ can be regarded as the inner-level path length to capture the variation of the optimal inner-level decisions as in [59]. Note that $V_{1,T}$ is a weaker regularity for the functional variation compared to absolute values used in single-level online optimization for dynamic regret [5, 66]. When the functions are static, these variations terms are simply 0. We are interested in the case where both $V_{1,T}$ and $H_{2,T}$ are $o(T)$ as in the literature of dynamic regret.

Based on Theorem 5.6, we can have the following theorem to characterize the regret of SOBOW.

**Theorem 5.7.** *Suppose that Assumptions 5.1-5.4 hold. Let $\lambda \leq \frac{1}{L_1}$, $\alpha \leq \frac{1}{L_1}$, $Q_{t+1} - Q_t \geq \frac{\log(1 - \frac{\alpha\mu_g}{2})}{2\log(1 - \lambda\mu_g)}$, $\eta \in (1 - \frac{\alpha\mu_g}{2}, 1)$, and $\beta \leq \min\{\frac{1}{4L_f}, \frac{\mu_g^2 L_f W(1 - \eta)(\eta - 1 + \alpha\mu_g/2)}{24L_1^4 G_2 \eta}\}$ where $L_f$ is the*

*smoothness parameter of the function $f_t(\cdot, y_t^*(\cdot))$. Then we can have*

$$BLR_w(T) \leq O\left(\frac{T}{\beta W} + \frac{V_{1,T}}{\beta} + H_{2,T}\right).$$

The value of $L_f$ can be found in the proof in Appendix. Note that the hypergradient estimation at the current round depends on all previous outer-level decisions as shown in Theorem 5.6. While these decisions may not be good at the early stage in OBO, choosing $\eta \in (1 - \frac{\alpha\mu_g}{2}, 1)$ would diminish their impact on the local regret. When the variations $V_{1,T}$ and $H_{2,T}$ are both $o(T)$, a sublinear bilevel local regret can be achieved for an appropriately selected window, e.g., $W = o(T)$ when $\eta = 1 - h(T)$, where $h(T) \to 0$ as $T \to \infty$. Note that the value $\beta$ does not change substantially since $W(1 - \eta)$ converges to 1. Particularly, when $\eta = 1 - o(\frac{1}{T})$, $W = \omega(T)$. In this case, we can have the smallest regret $O\left(\frac{V_{1,T}}{\beta} + H_{2,T}\right)$ that only depends on the function variations in OBO.

## 6 Experiments

In this section, we conduct experiments in multiple domains to corroborate the utility of the OBO framework and the effectiveness of SOBOW.

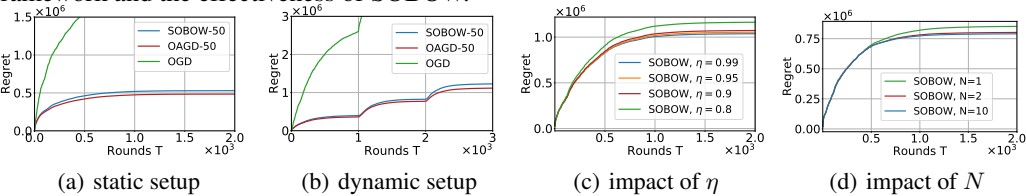

(a) static setup    (b) dynamic setup    (c) impact of $\eta$    (d) impact of $N$

Figure 1: Evaluation for online HR. As shown in subfigures (a) and (b), SOBOW performs similarly to OAGD and significantly outperforms OGD in both static and dynamic setups. Subfigure (c) shows the performance of SOBOW under different values of the averaging parameter $\eta$ for online HR. Better performance is achieved as $\eta \to 1$. Subfigure (d) shows the performance of SOBOW under different values of inner steps $N$ when the data stream contains two data points (one in $X_t^g$ and one in $X_t^f$). The performance saturates at $N = 2$.

Specifically, we compare our algorithm SOBOW with the following baseline methods: (1) **OAGD** [59], which is the only method for OBO in the literature; (2) **OGD**, a natural method which updates the outer-level decision by using the current hypergradient estimation only without any window averaging. Intuitively, *OGD is not only a special case of OAGD when the information of previous functions is not available*, but also a direct application of offline bilevel optimization, e.g., AID-based method [32]. We also denote SOBOW-$K$/OAGD-$K$ as SOBOW and OAGD with window size $K$, respectively. And we evaluate the regret using the definition in Equation (2). We also compare the performance using the regret in [59] in Appendix where similar results can be observed.

**Online Hyper-representation Learning** Representation learning [13, 17] seeks to extract good representations of the data. The learnt representation mapping can be used in downstream tasks to facilitate the learning of task specific model parameters. This formulation is typically encountered in a multi-task setup, where $\Lambda$ captures the common representation extracted for multiple tasks and $w$ defines the task-specific model parameters. When the data/task arrives in an online manner, the hyper-representation needs to be continuously adapted to incorporate the new knowledge.

Following [17], we study online hyper-representation learning (Online HR) with linear models. Specifically, at each round $t$, the agent applies the hyper-representation $\Lambda_t \in \mathbb{R}^{p \times d}$ and the linear model prediction $w_t \in \mathbb{R}^d$, and then receives small minibatches $(X_t^f, Y_t^f)$ and $(X_t^g, Y_t^g)$. Based on $\Lambda_t$ and $(X_t^g, Y_t^g)$, the agent updates her linear model prediction $w_{t+1}$ as an estimation of $w^*(\Lambda_t) = \arg\min_{w \in \mathbb{R}^d} g_t(\Lambda_t, w) := \|X_t^g \Lambda_t w - Y_t^g\|^2 + \frac{\gamma}{2}\|w\|^2$. Based on the estimation $w_{t+1}$ and $(X_t^f, Y_t^f)$, the agent further updates her decision $\Lambda_{t+1}$ about the hyper-representation to minimize the loss $f_t(\Lambda, w_t^*(\Lambda)) := \|X_t^f \Lambda w_t^*(\Lambda) - Y_t^f\|^2$. In our experiments, we consider synthetic data generated as in [17] and explore two distinct settings: (i) a static setup where the underlying model generating the minibatches is fixed; and (ii) a staged dynamic setup where the model changes after some steps.

As shown in Figure 1(a) and Figure 1(b), SOBOW achieves comparable regret with OAGD in both static and dynamic setups, without the need of knowing previous functions. In terms of the running

Table 2: **Left:** Comparison for static online HO. We report accuracy and loss on a separate test split after 12000 steps. **Right:** Comparison for dynamic online HO. We report accuracy on a separate test split at the end of stream with corruption level 20% and 30%. Each level lasts for 4000 steps.

| Method | Accuracy (%) | Test Loss | Time (s) |
|---|---|---|---|
| SOBOW-4 | 65.87 | 1.287 | 899 |
| OAGD-4 | 65.96 | 1.285 | 2304 |
| SOBOW-50 | 66.32 | 1.28 | 1188 |
| OAGD-50 | 66.44 | 1.273 | 20161 |

| Method | End 20% stream | End 30% stream | Time (s) |
|---|---|---|---|
| SOBOW-4 | 58.39 | 59.70 | 1198 |
| OAGD-4 | 62.61 | 59.26 | 3072 |

time for 5000 steps with $K = 50$, SOBOW takes 11 seconds, OAGD takes 228 seconds and OGD takes 7 seconds. Therefore, SOBOW is much more computationally efficient compared to OAGD, because SOBOW does not need to re-evaluate the previous functions on the current model at each round. On the other hand, *SOBOW performs substantially better than OGD (i.e., OAGD when previous functions are not available) with similar running time*. These results not only demonstrate the usefulness of SOBOW when the previous functions are not available, but also corroborate the benefit of window-averaged outer-level decision update by leveraging the historical hypergradient estimations in OBO. Figure 1(c) shows the performance of SOBOW under different values of the averaging parameter $\eta$. The performance is better as $\eta \to 1$, which is also consistent with our theoretical results. Figure 1(d) indicates that a small number of updates for the inner-level variable is indeed enough for online HR.

**Online Hyperparameter Optimization** The goal of hyperparameter optimization (HO) [13, 17] is to search for the best values of hyperparameters $\lambda$, which seeks to minimize the validation loss of the learnt model parameters $w$ and is usually done offline. However, in online applications where the data distribution can dynamically change, e.g., the unusual traffic patterns in online traffic time series prediction problem [63], keeping the hyperparameters static could lead to sub-optimal performance. Therefore, the hyperparameters should be continuously updated together with the model parameters in an online manner.

Specifically, at each online round $t$, the agent applies the hyperparameters $\lambda_t$ and the model $w_t$, and then receives a small dataset $\mathcal{D}_t = \{\mathcal{D}_t^{\mathrm{tr}}, \mathcal{D}_t^{\mathrm{val}}\}$ composed of a training subset $\mathcal{D}_t^{\mathrm{tr}}$ and a validation subset $\mathcal{D}_t^{\mathrm{val}}$. Based on $\lambda_t$ and $\mathcal{D}_t^{\mathrm{tr}}$, the agent first updates her model prediction $w_{t+1}$ as an estimation of $w_t^*(\lambda_t) := \arg\min_w \mathcal{L}_t^{\mathrm{tr}}(\lambda_t, w)$, where $\mathcal{L}_t^{\mathrm{tr}}(\lambda, w) := \frac{1}{|\mathcal{D}_t^{\mathrm{tr}}|} \sum_{\zeta \in \mathcal{D}_t^{\mathrm{tr}}} \mathcal{L}(w; \zeta) + \Omega(\lambda, w)$, $\mathcal{L}(w, \zeta)$ is a cost function computed on data point $\zeta$ with prediction model $w$, and $\Omega(w, \lambda)$ is a regularizer. Based on the model prediction $w_{t+1}$ and $\mathcal{D}_t^{\mathrm{tr}}$, the agent updates the hyperparameters $\lambda_{t+1}$ to minimize the validation loss $\mathcal{L}_t^{\mathrm{val}}(\lambda, w_t^*(\lambda)) := \frac{1}{|\mathcal{D}_t^{\mathrm{val}}|} \sum_{\xi \in \mathcal{D}_t^{\mathrm{val}}} \mathcal{L}(w_t^*(\lambda); \xi)$.

We consider an online classification setting on the 20 Newsgroup dataset, where the classifier is modeled by an affine transformation and we use the cross-entropy loss as the losscost function. For $\Omega(\lambda, w)$, we use one $\ell_2$-regularization parameter for each row of the transformation matrix in $w$, so that we have one regularization parameter for each data feature (i.e., $|\lambda|$ is given by the dimension of the data). We remove all news headers in the 20 Newsgroup dataset and pre-process the dataset so as to have data feature vectors of dimension $d = 99238$. In our implementations, we approximate the hypergradient using implicit differentiation with the fixed point method [17]. We consider two different setups: (i) a static setup where the agent receives a stream of clean data batches $\{\mathcal{D}_t\}_t$; (ii) a dynamic setting in which the agent receives a stream of corrupted batches $\{\mathcal{D}_t\}_t$, where the corruption level changes after some time steps. For both setups the batchsize is fixed to 16. For the dynamic setting we consider four different corruption levels $\{5\%, 10\%, 20\%, 30\%\}$ and also optimize the learning rate as an additional hyperparameter.

We evaluate the testing accuracy for SOBOW and OAGD in Table 2 for both static (Left) and dynamic (Right) setups. It can be seen that compared to OAGD, SOBOW achieves similar accuracy but with a much shorter running time. When the window size increases in Table 2, performance of both SOBOW and OAGD increases and the computational advantage of SOBOW becomes more significant. In particular, SOBOW runs around 20 times faster than OAGD when the window size is 50.

# 7 Conclusions and Discussion

In this work, we study non-convex bilevel optimization where the functions can be time-varying and the agent continuously updates the decisions with online streaming data. We proposed a single-loop

online bilevel optimizer with window averaging (SOBOW) to handle the function variations and the unavailability of the true hypergradients in OBO. Compared to existing algorithms, SOBOW is computationally efficient and does not require previous function information. We next developed a novel analytical technique to tackle the unique challenges in OBO and showed that SOBOW can achieve a sublinear bilevel local regret. Extensive experiments justified the effectiveness of SOBOW. We also discuss the potential applications of the OBO framework in online meta-learning and online adversarial training (see Appendix).

*Limitation and future directions* The study of online bilevel optimization is still in a very early stage, and much of this new framework still remains under-explored and not well understood. We started with the second-order approach for hypergradient estimation, which is less scalable. One future direction is to leverage the recently developed first order approaches for hypergradient estimation. Another limitation is that we assume that the inner-level objective function is strongly convex. In the future, we will investigate the convex and even non-convex case.

## Acknowledgments

This work has been supported in part by NSF grants NSF AI Institute (AI-EDGE) CNS-2112471, CNS-2106933, CNS-2106932, CNS-2312836, CNS-1955535, CNS-1901057, ECCS-2113860, DMS-2134145, and 2311274, by Army Research Office under Grant W911NF-21-1-0244, and was sponsored by the Army Research Laboratory and was accomplished under Cooperative Agreement Number W911NF-23-2-0225. The views and conclusions contained in this document are those of the authors and should not be interpreted as representing the official policies, either expressed or implied, of the Army Research Laboratory or the U.S. Government. The U.S. Government is authorized to reproduce and distribute reprints for Government purposes notwithstanding any copyright notation herein.

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

# Appendix

## A   Discussion about practical applications of OBO

(1) In the traffic flow prediction problem [63], since the data collected by the traffic sensors arrive at the controller continuously and frequently, the hyperparameters need to be optimized (in the outer level) quickly in an online manner in order to guarantee the performance of the prediction model (in the inner level). Keeping the hyperparameters static for the prediction model may result in sub-optimal performance, because the distribution of the traffic flow can change gradually. Further, the controller may not have global information of the traffic flows, because it is exorbitantly expensive to deploy sensors to cover all traffic flows.

(2) In the wireless network control problem [37], the controller allocates resources, e.g., wireless channel bandwidth, to the users, where each user can have its own utility function depending on the wireless channel conditions given the allocated resources. Since wireless channels are usually time-varying, the controller has to continuously update the resource allocation quickly to maximize the network performance. The rate allocation decisions (deciding what packet rate a user transmits at any given time) and scheduling decisions (which users transmit) are done at a fast time-scale (in the inner level), while determining the utility functions of the users could be done at a slower time scale (in the outer level). Further these decisions need to be made in a distributed fashion, so under local knowledge of the channel conditions and interference levels.

In these applications, the current offline bilevel optimization framework cannot be directly applied because of the streaming data, the time-varying functions and possibly the limited information about the system. In contrast, online bilevel optimization has great potential for these online applications. Moreover, the decision making in these online applications also needs to be fast and efficient without the need of knowing all previous functions. Thus, the regret captures the performance of the learning model over the sequential process rather than just the performance of a final output model in the offline setting. How to design such algorithms with a sublinear regret guarantee is very important for making online bilevel optimization more practical in real applications.

## B   Applications of OBO in meta-learning and adversarial training

**Online meta-learning**   In online meta-learning, learning tasks arrive one at a time, and the agent aims to learn a good meta-model $\theta$ based on the past tasks in a sequential manner, which can be quickly adapted to a good task-specific model $\phi$ for the current task. Specifically, each task t has a training dataset $\mathcal{D}_t^{tr}$ and a testing dataset $\mathcal{D}_t^{te}$, and given a meta- model $\theta$, the optimal task model is defined as

$$\phi_t^*(\theta) \in \arg\min_{\phi} g_t(\theta, \phi) = \mathcal{L}_t(\phi, \mathcal{D}_t^{tr}) + \lambda\|\theta - \phi\|^2.$$

In the OBO framework of online meta-learning, the meta-model $\theta_t$ is the outer-level decision variable at round $t$, and the task model $\phi_t$ is the inner-level decision variable. At round $t$, the task model $\phi_{t+1}$ is first obtained based on $g_t(\theta_t, \phi)$ as an estimation of $\phi_t^*(\theta_t)$; and then given $\phi_{t+1}$, the meta-model will be further updated w.r.t.

$$f_t(\theta, \phi_{t+1}) = \mathcal{L}_t(\phi_{t+1}, \mathcal{D}_t^{te}) + \lambda\|\theta - \phi_{t+1}\|^2.$$

**Online adversarial training**   Adversarial training [46, 60, 65] is usually formulated as a min-max optimization problem. The defender learns a robust model to minimize the worst-case training loss against an attacker, where the attacker aims to maximize the loss by perturbing the training data. A static setting is often considered with full access to the target dataset at all times. Nevertheless, many real-world applications involve streaming data that arrive in an online manner, e.g., the financial markets or real-time sensor networks. A continuously robust model update is more desirable in these applications against potential attacks on the streaming data.

This online adversarial training problem can also be addressed by the OBO framework. Here the model parameters $\theta$ is the outer-level decision variable and the adversarial perturbations $\delta$ to the data point $(x, y)$ is the inner-level decision variable. At each time $t$, the defender updates the estimate $\delta_{t+1}$ of the worst adversarial perturbations, given her knowledge about the inner-level objection function $g_t(\theta_t, \delta_t) = -\mathcal{L}(\theta_t, x_t + \delta, y_t)$ subject to the perturbation constraint. Based

on $\delta_{t+1}$, the defender updates the model $\theta_{t+1}$ robustly w.r.t. the outer-level objective function $f_t(\theta, \delta_{t+1}) = \mathcal{L}(\theta, x_t + \delta_{t+1}, y_t)$.

## C  Experimental details

We use a grid of values between $10^{-5}$ and $10$ to set the stepsizes and did not find the algorithm particularly sensitive to them for the experiments considered. For example, we achieve best performance by setting both the inner and outer stepsizes to $10^{-4}$ for the online hyper-representation learning experiments and small values around that scale yield the same performance. For the dynamic OHO experiments, only the outer step size is set manually to $0.01$. The inner step size is optimized along with the other regularization hyperparameters.

OAGD needs to store the previous objective functions, which requires all the previous data points to compute the function values and additional resources to store the knowledge of the function structures. In contrast, our method only stores the previous hypergradient estimates averaged over previous data points. For example, when $n$ data points are used to evaluate the function at each round, the memory requirement for OAGD is $O(nK)$, compared to $O(K)$ for our method. Here, $K$ is the window size. Therefore, the memory cost of our method can be lower, especially when the number of data points is large.

## D  Comparison between the regret definitions

For a clear comparison, we restate the definitions of bilevel local regret in our work and OAGD here:

Our definition:

$$BLR = \sum_{t=1}^{T} \left\| \frac{1}{W} \sum_{i=1}^{K-1} \eta^i \nabla f_{t-i}(x_{t-i}, y_{t-i}^*(x_{t-i})) \right\|^2.$$

In OAGD:

$$BLR = \sum_{t=1}^{T} \left\| \frac{1}{W} \sum_{i=1}^{K-1} \eta^i \nabla f_{t-i}(x_t, y_t^*(x_t)) \right\|^2.$$

The key difference is that we evaluate the past loss $f_{t-i}$ using the variable updates $x_{t-i}$ and $y_{t-i}^*(x_{t-i})$ at exactly same time $t-i$, while in OAGD the past loss $f_{t-i}$ is evaluated using the most recent updates $x_t$ and $y_t^*(x_t)$. As shown in [2], the static regret in OAGD can cause problems for time-varying loss functions. Intuitively, evaluating the objective at time slot $i$ using variable updates at different time slot $j$ can be misleading, because it does not properly characterize the online learning performance of the model update at time slot $i$, especially when the objective functions vary a lot.

In Figure 2 we compare our algorithm and OAGD using the regret notion proposed in OAGD. The results show that our algorithm still achieves similar regret performance compared to OAGD, but with a much shorter runtime.

## E  Additional Results

Using path-length regularization to capture the variation of optimal decision variables is very common in the literature of dynamic online learning, e.g., [68, 48, 61, 64, 66]. Note that because this variation of optimal decision variables is not controllable, we do not use this term in the design of the algorithm. Rather, the variation term is only used in the theoretical analysis to understand which factors in the system lead to a tighter bound on the regret. However, we mention here that it is also possible to explicitly analyze the regret in terms of the function variations directly.

**Theorem E.1.** *Suppose that Assumptions 5.1-5.4 hold. Let $V_g = \sum_{t=1}^{T} \sup |g_{t+1}(x, y) - g_t(x, y)|$ and $V_f = \sum_{t=1}^{T} \sup[f_{t+1}(x, y) - f_t(x, y)]$. Under the same conditions on $\lambda$, $\alpha$, $Q_t$, $\eta$ and $\beta$ with Theorem 5.7, we can have $BLR_w(T) \leq O\left( \frac{T}{\beta W} + \frac{V_f}{\beta} + V_g + \frac{\sqrt{T V_g}}{\beta} \right)$.*

The main proof idea is as follows:

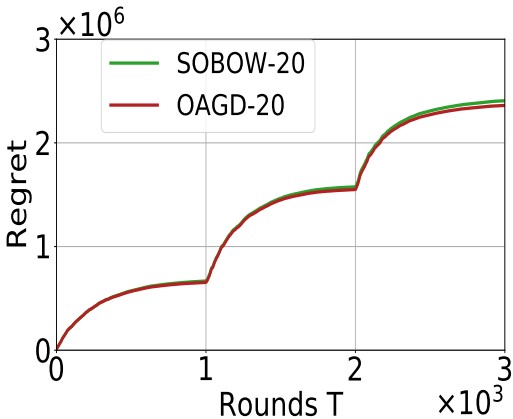

Figure 2: Comparison between our SOBOW and OAGD using the regret notion proposed in [59] on the dynamic online HR problem.

(1) For the term $H_{2,T}$, based on the strong convexity of $g_t$, we can show that $\|y_{t+1}^*(x) - y_t^*(x)\|^2 \leq \frac{2}{\mu_g} \sup |g_{t+1}(x,y) - g_t(x,y)|$;

(2) For the term $V_{1,T}$, we can show that $f_{t+1}(x, y_{t+1}^*(x)) - f_t(x, y_t^*(x)) \leq L_0 \|y_{t+1}^*(x) - y_t^*(x)\| + \sup[f_{t+1}(x,y) - f_t(x,y)]$, such that $\sum_{t=1}^{T}[F_{t,\eta}(x_t, y_t^*(x_t)) - F_{t,\eta}(x_{t+1}, y_t^*(x_{t+1}))]$ in Lemma H.3 can be upper bounded by $\frac{2MT}{W} + L_0\sqrt{\frac{2T}{\mu_g}}\sqrt{\sum_{t=1}^{T} \sup |g_{t+1}(x,y) - g_t(x,y)|} + \sum_{t=1}^{T} \sup[f_{t+1}(x,y) - f_t(x,y)]$;

(3) Based on the above, if we denote $V_g = \sum_{t=1}^{T} \sup |g_{t+1}(x,y) - g_t(x,y)|$ and $V_f = \sum_{t=1}^{T} \sup[f_{t+1}(x,y) - f_t(x,y)]$ to capture the function variations, we can have the overall regret as $O\left(\frac{T}{\beta W} + \frac{V_f}{\beta} + V_g + \frac{\sqrt{TV_g}}{\beta}\right)$. In this case, a sublinear regret will be achieved if both $V_g$ and $V_f$ are $o(T)$ for suitably selected $W$. As mentioned in our previous response, the condition on the variation of $y_t^*(x)$ is weaker compared to the condition on the variation of $g_t$ in order to achieve a small regret. For example, suppose $W = \omega(T)$ and the function variation of $f_t$ is very small, to achieve a regret of $O(T^{3/4})$, $V_{1,T} = O(T^{3/4})$ is sufficient, while we need a stricter condition on the function variation of $g_t$, i.e., $V_g = O(T^{1/2})$.

## F  Proof of Lemma 5.5

To prove Lemma 5.5, we first have the following lemma about $v_t^*$:

**Lemma F.1.** *Suppose that Assumption 5.2, Assumption 5.3 and Assumption 5.4 hold. We can have the following upper bound on $\|v_t^*\|$:*

$$\|v_t^*\| \leq \frac{\rho\mu_g + DL_1^2 + DL_1\mu_g}{\mu_g^2} \triangleq M_v.$$

*Proof.* Based on Lemma 2.2 in [14], it can be shown that $y_t^*(x)$ is $\frac{L_1}{\mu_g}$-Lipschitz continuous in $x$, i.e.,

$$\|y_t^*(x) - y_t^*(x')\| \leq \frac{L_1}{\mu_g}\|x - x'\| \tag{8}$$

for any $x$ and $x'$. According to Assumption 5.4, let $\hat{x} \in \mathcal{X}$ such that $\|\nabla_y f_t(\hat{x}, y_t^*(\hat{x}))\| \leq \rho$. Then it follows that

$$\|\nabla_y f_t(x_t, y_t^*(x_t))\| = \|\nabla_y f_t(\hat{x}, y_t^*(\hat{x})) + \nabla_y f_t(x_t, y_t^*(x_t)) - \nabla_y f_t(\hat{x}, y_t^*(\hat{x}))\|$$

$$\leq \|\nabla_y f_t(\hat{x}, y_t^*(\hat{x}))\| + \|\nabla_y f_t(x_t, y_t^*(x_t)) - \nabla_y f_t(\hat{x}, y_t^*(\hat{x}))\|$$

$$\leq \rho + \left(L_1 + \frac{L_1^2}{\mu_g}\right)\|x_t - \hat{x}\|$$

$$\leq \frac{\rho\mu_g + DL_1^2 + DL_1\mu_g}{\mu_g}.$$

Therefore,

$$\|v_t^*\| = \|(\nabla_y^2 g_t(x_t, y_t^*(x_t)))^{-1}\nabla_y f_t(x_t, y_t^*(x_t))\|$$

$$\leq \|(\nabla_y^2 g_t(x_t, y_t^*(x_t)))^{-1}\|\|\nabla_y f_t(x_t, y_t^*(x_t))\|$$

$$\leq \frac{1}{\mu_g}\|\nabla_y f_t(x_t, y_t^*(x_t))\|$$

$$\leq \frac{\rho\mu_g + DL_1^2 + DL_1\mu_g}{\mu_g^2}.$$

$\square$

Based on Lemma 1 in [30], we can have that

$$\|v_t^Q - v_t^*\|$$

$$\leq \left[Q_t(1 - \lambda\mu_g)^{Q_t-1}L_2\lambda M_v + \frac{1 - (1 - \lambda\mu_g)^{Q_t}(1 + \lambda Q_t\mu_g)}{\mu_g}L_2 M_v\right]\|y_t^* - y_{t+1}\|$$

$$+ (1 - (1 - \lambda\mu_g)^{Q_t})\frac{L_1}{\mu_g}\|y_t^* - y_{t+1}\| + (1 - \lambda\mu_g)^{Q_t}\|v_t^0 - v_t^*\|$$

$$\leq \left[\frac{1}{1 - \lambda\mu_g}\frac{Q_t}{1 + Q_t\log\frac{1}{1-\lambda\mu_g}}L_2\lambda M_v + \frac{1}{\mu_g}L_2 M_v + \frac{L_1}{\mu_g}\right]\|y_t^* - y_{t+1}\| + (1 - \lambda\mu_g)^{Q_t}\|v_t^0 - v_t^*\|$$

$$= \left(\frac{L_2\lambda M_v}{(1 - \lambda\mu_g)\log\frac{1}{1-\lambda\mu_g}} + \frac{L_2 M_v + L_1}{\mu_g}\right)\|y_t^* - y_{t+1}\| + (1 - \lambda\mu_g)^{Q_t}\|v_t^0 - v_t^*\|$$

$$\triangleq C_Q\|y_t^* - y_{t+1}\| + (1 - \lambda\mu_g)^{Q_t}\|v^0 - v_t^*\|$$

where $C_Q = \frac{L_2\lambda M_v}{(1-\lambda\mu_g)\log\frac{1}{1-\lambda\mu_g}} + \frac{L_2 M_v + L_1}{\mu_g}$. By using the Young's inequality and Lemma F.1, it follows that

$$\|v_t^Q - v_t^*\|^2 \leq \left(1 + \frac{1}{\lambda\mu_g}\right)C_Q^2\|y_t^* - y_{t+1}\|^2 + (1 + \lambda\mu_g)(1 - \lambda\mu_g)^{2Q_t}\|v^0 - v_t^*\|^2$$

$$\leq \left(1 + \frac{1}{\lambda\mu_g}\right)C_Q^2\|y_t^* - y_{t+1}\|^2 + (1 + \lambda\mu_g)(1 - \lambda\mu_g)^{2Q_t}(2\|v^0\|^2 + 2M_v)$$

$$\triangleq c_2\|y_t^* - y_{t+1}\|^2 + \epsilon_t^2$$

where $c_2 = \left(1 + \frac{1}{\lambda\mu_g}\right)C_Q^2$, and $\epsilon_t^2 = (1 + \lambda\mu_g)(1 - \lambda\mu_g)^{2Q_t}(2\|v^0\|^2 + 2M_v)$.

## G Proof of Theorem 5.6

Based on Equation (3) and Equation (4), it follows that

$$\|\nabla f_t(x_t, y_t^*(x_t)) - \widehat{\nabla}f_t(x_t, y_{t+1})\|^2$$

$$= \|\nabla_x f_t(x_t, y_t^*(x_t)) - \nabla_x\nabla_y g_t(x_t, y_t^*(x_t))v_t^* - \nabla_x f_t(x_t, y_{t+1}) + \nabla_x\nabla_y g_t(x_t, y_{t+1})v_v^Q\|^2$$

$$
\begin{aligned}
=&\|\nabla_x f_t(x_t, y_t^*(x_t)) - \nabla_x f_t(x_t, y_{t+1}) + \nabla_x \nabla_y g_t(x_t, y_{t+1}) v_t^Q - \nabla_x \nabla_y g_t(x_t, y_{t+1}) v_t^* \\
& + \nabla_x \nabla_y g_t(x_t, y_{t+1}) v_t^* - \nabla_x \nabla_y g_t(x_t, y_t^*(x_t)) v_t^*\|^2 \\
\leq & 3\|\nabla_x f_t(x_t, y_{t+1}) - \nabla_x f_t(x_t, y_t^*(x_t))\|^2 + 3\|\nabla_x \nabla_y g_t(x_t, y_{t+1})\|^2 \|v_t^* - v_t^Q\|^2 \\
& + 3\|\nabla_x \nabla_y g_t(x_t, y_t^*(x_t)) - \nabla_x \nabla_y g_t(x_t, y_{t+1})\|^2 \|v_t^*\|^2 \\
\overset{(a)}{\leq} & 3L_1^2 \|y_{t+1} - y_t^*(x_t)\|^2 + 3L_1^2 \|v_t^* - v_t^Q\|^2 + 3L_2^2 \|y_{t+1} - y_t^*(x_t)\|^2 \|v_t^*\|^2 \\
\overset{(b)}{\leq} & \left( 3L_1^2 + \frac{3L_2^2(\rho \mu_g + DL_1^2 + DL_1 \mu_g)^2}{\mu_g^4} \right) \|y_{t+1} - y_t^*(x_t)\|^2 + 3L_1^2 \|v_t^* - v_t^Q\|^2 \\
\overset{(c)}{\leq} & \left( 3L_1^2(1 + c_2) + \frac{3L_2^2(\rho \mu_g + DL_1^2 + DL_1 \mu_g)^2}{\mu_g^4} \right) \|y_{t+1} - y_t^*(x_t)\|^2 + 3L_1^2 \epsilon_t^2 \\
=& 3L_1^2 \left[ \left( 1 + c_2 + \frac{L_2^2(\rho \mu_g + DL_1^2 + DL_1 \mu_g)^2}{L_1^2 \mu_g^4} \right) \|y_{t+1} - y_t^*(x_t)\|^2 + \epsilon_t^2 \right] \quad\quad (9)
\end{aligned}
$$

where (a) is true because of Assumption 5.2, (b) is true because of Lemma F.1 and (c) holds due to Lemma 5.5.

Next, based on Assumption 5.1 and Assumption 5.2, for any $t$ and $\alpha < \frac{1}{L_1}$ we can have

$$
\begin{aligned}
& \|y_{t+1} - y_t^*(x_t)\|^2 \\
\leq & (1 - \alpha \mu_g) \|y_t - y_t^*(x_t)\|^2 \\
= & (1 - \alpha \mu_g) \|y_t - y_{t-1}^*(x_{t-1}) + y_{t-1}^*(x_{t-1}) - y_t^*(x_t)\|^2 \\
\overset{(a)}{\leq} & (1 + \lambda)(1 - \alpha \mu_g) \|y_t - y_{t-1}^*(x_{t-1})\|^2 + (1 + \frac{1}{\lambda})(1 - \alpha \mu_g) \|y_{t-1}^*(x_{t-1}) - y_t^*(x_t)\|^2 \\
\leq & (1 + \lambda)(1 - \alpha \mu_g) \|y_t - y_{t-1}^*(x_{t-1})\|^2 + 2(1 + \frac{1}{\lambda})(1 - \alpha \mu_g) \|y_{t-1}^*(x_{t-1}) - y_t^*(x_{t-1})\|^2 \\
& + 2(1 + \frac{1}{\lambda})(1 - \alpha \mu_g) \|y_t^*(x_{t-1}) - y_t^*(x_t)\|^2 \\
\overset{(b)}{\leq} & (1 + \lambda)(1 - \alpha \mu_g) \|y_t - y_{t-1}^*(x_{t-1})\|^2 + 2(1 + \frac{1}{\lambda})(1 - \alpha \mu_g) \|y_{t-1}^*(x_{t-1}) - y_t^*(x_{t-1})\|^2 \\
& + \frac{2L_1^2}{\mu_g^2}(1 + \frac{1}{\lambda})(1 - \alpha \mu_g) \|x_{t-1} - x_t\|^2,
\end{aligned}
$$

where (a) is based on the Young's inequality and (b) is due to Equation (8).

For $\lambda = \frac{\alpha \mu_g}{2}$, it follows that

$$
\begin{aligned}
(1 + \lambda)(1 - \alpha \mu_g) =& (1 + \frac{\alpha \mu_g}{2})(1 - \alpha \mu_g) \\
=& 1 - \frac{\alpha \mu_g}{2} - \frac{\alpha^2 \mu_g^2}{2} \\
<& 1 - \frac{\alpha \mu_g}{2}.
\end{aligned}
$$

Let $G_1 = 1 + c_2 + \frac{L_2^2(\rho \mu_g + DL_1^2 + DL_1 \mu_g)^2}{L_1^2 \mu_g^4}$ and $\delta_t = G_1 \|y_{t+1} - y_t^*(x_t)\|^2 + \epsilon_t^2$. Based on Lemma 5.5, we can have that

$$
\begin{aligned}
\delta_t = & G_1 \|y_{t+1} - y_t^*(x_t)\|^2 + \epsilon_t^2 \\
\leq & G_1(1 + \lambda)(1 - \alpha \mu_g) \|y_t - y_{t-1}^*(x_{t-1})\|^2 + 2G_1(1 + \frac{1}{\lambda})(1 - \alpha \mu_g) \|y_{t-1}^*(x_{t-1}) - y_t^*(x_{t-1})\|^2 \\
& + \frac{2L_1^2 G_1}{\mu_g^2}(1 + \frac{1}{\lambda})(1 - \alpha \mu_g) \|x_{t-1} - x_t\|^2 + \epsilon_t^2 \\
\leq & G_1 \left( 1 - \frac{\alpha \mu_g}{2} \right) \|y_t - y_{t-1}^*(x_{t-1})\|^2 + G_2 \|y_{t-1}^*(x_{t-1}) - y_t^*(x_{t-1})\|^2
\end{aligned}
$$

$$+ \frac{L_1^2 G_2}{\mu_g^2} \|x_{t-1} - x_t\|^2 + \left(1 - \frac{\alpha\mu_g}{2}\right)\epsilon_{t-1}^2$$

$$= \left(1 - \frac{\alpha\mu_g}{2}\right)\delta_{t-1} + G_2\|y_{t-1}^*(x_{t-1}) - y_t^*(x_{t-1})\|^2 + \frac{L_1^2 G_2}{\mu_g^2}\|x_{t-1} - x_t\|^2$$

$$\leq \left(1 - \frac{\alpha\mu_g}{2}\right)^{t-1}\delta_1 + G_2\sum_{j=0}^{t-2}\left(1 - \frac{\alpha\mu_g}{2}\right)^j \|y_{t-1-j}^*(x_{t-1-j}) - y_{t-j}^*(x_{t-1-j})\|^2$$

$$+ \frac{2L_1^2 G_2}{\mu_g^2}\sum_{j=0}^{t-2}\left(1 - \frac{\alpha\mu_g}{2}\right)^j \|x_{t-1-j} - x_{t-j}\|^2 \tag{10}$$

where $G_2 = 2G_1(1 + \frac{2}{\alpha\mu_g})(1 - \alpha\mu_g)$ and $\delta_1 = \left(1 + c_2 + \frac{L_2^2(\rho\mu_g + DL_1^2 + DL_1\mu_g)^2}{L_1^2\mu_g^4}\right)\|y_2 - y_1^*(x_1)\|^2 + \epsilon_1^2$.

By substituting Equation (10) back into Equation (9), we can obtain that

$$\|\nabla f_t(x_t, y_t^*(x_t)) - \widehat{\nabla}f_t(x_t, y_{t+1})\|^2$$

$$\leq 3L_1^2 \Bigg\{ \left(1 - \frac{\alpha\mu_g}{2}\right)^{t-1}\delta_1 + G_2\sum_{j=0}^{t-2}\left(1 - \frac{\alpha\mu_g}{2}\right)^j \|y_{t-1-j}^*(x_{t-1-j}) - y_{t-j}^*(x_{t-1-j})\|^2$$

$$+ \frac{2L_1^2 G_2}{\mu_g^2}\sum_{j=0}^{t-2}\left(1 - \frac{\alpha\mu_g}{2}\right)^j \|x_{t-1-j} - x_{t-j}\|^2 \Bigg\}.$$

## H Proof of Theorem 5.7

Based on Lemma 2.2 in [14], we can have the following lemma to characterize the smoothness of the function $f_t(x, y_t^*(x))$ w.r.t. $x$ for any $t \in [1, T]$.

**Lemma H.1.** *Suppose that Assumption 5.1 and Assumption 5.2 hold. Then for any $t \in [1, T]$, the function $f_t(x, y_t^*(x))$ is $L_f$-smooth, i.e., for any $x$ and $x'$,*

$$\|\nabla f_t(x, y_t^*(x)) - \nabla f_t(x', y_t^*(x'))\| \leq L_f\|x - x'\|,$$

*where the constant $L_f$ is given by*

$$L_f = L_1 + \frac{2L_1^2 + L_0^2 L_2}{\mu_g} + \frac{L_1^3 + 2L_0 L_1 L_2}{\mu_g^2} + \frac{L_0 L_1^2 L_2}{\mu_g^3}.$$

For any $t \in [1, T]$ we first define

$$F_{t,\eta}(x_{t+1}, y_t^*(x_{t+1})) = \frac{1}{W}\sum_{i=0}^{K-1}\eta^i f_{t-i}(x_{t+1-i}, y_{t-i}^*(x_{t+1-i})).$$

Based on Assumption 5.2, we can show the smoothness of $F_{t,\eta}(x, y_t^*(x))$ w.r.t. $x$.

**Lemma H.2.** *Suppose Assumption 5.2 holds. Then the following holds for function $F_{t,\eta}(x_{t+1}, y_t^*(x_{t+1}))$:*

$$F_{t,\eta}(x_{t+1}, y_t^*(x_{t+1})) - F_{t,\eta}(x_t, y_t^*(x_t)) \leq \langle \nabla F_{t,\eta}(x_t, y_t^*(x_t)), x_{t+1} - x_t \rangle + \frac{L_f}{2}\|x_{t+1} - x_t\|^2.$$

*Proof.* For any $x$ and $x'$, we can know that

$$F_{t,\eta}(x_{t+1}, y_t^*(x_{t+1})) - F_{t,\eta}(x_t, y_t^*(x_t))$$

$$= \frac{1}{W}\sum_{i=0}^{K-1}\eta^i f_{t-i}(x_{t+1-i}, y_{t-i}^*(x_{t+1-i})) - \frac{1}{W}\sum_{i=0}^{K-1}\eta^i f_{t-i}(x_{t-i}, y_{t-i}^*(x_{t-i}))$$

$$= \frac{1}{W}\sum_{i=0}^{K-1}\eta^i[f_{t-i}(x_{t+1-i}, y_{t-i}^*(x_{t+1-i})) - f_{t-i}(x_{t-i}, y_{t-i}^*(x_{t-i}))]$$

$$\overset{(a)}{\leq} \frac{1}{W} \sum_{i=0}^{K-1} \eta^i \left[ \langle \nabla f_{t-i}(x_{t-i}, y_{t-i}^*(x_{t-i})), x_{t+1} - x_t \rangle + \frac{L_f}{2} \|x_{t+1} - x_t\|^2 \right]$$

$$= \left\langle \frac{1}{W} \sum_{i=0}^{K-1} \eta^i \nabla f_{t-i}(x_{t-i}, y_{t-i}^*(x_{t-i})), x_{t+1} - x_t \right\rangle + \frac{L_f}{2} \|x_{t+1} - x_t\|^2$$

$$= \langle \nabla F_{t,\eta}(x_t, y_t^*(x_t)), x_{t+1} - x_t \rangle + \frac{L_f}{2} \|x_{t+1} - x_t\|^2$$

where (a) holds because of the smoothness of function $f_t(x, y_t^*(x))$ w.r.t. $x$.

$\square$

Next, based on Lemma H.2, we can have that

$$F_{t,\eta}(x_{t+1}, y_t^*(x_{t+1})) - F_{t,\eta}(x_t, y_t^*(x_t))$$

$$\leq \langle \nabla F_{t,\eta}(x_t, y_t^*(x_t)), x_{t+1} - x_t \rangle + \frac{L_f}{2} \|x_{t+1} - x_t\|^2$$

$$\leq -\beta \langle \nabla F_{t,\eta}(x_t, y_t^*(x_t)), \widehat{\nabla} F_{t,\eta}(x_t, y_{t+1}) \rangle + \frac{\beta^2 L_f}{2} \|\widehat{\nabla} F_{t,\eta}(x_t, y_{t+1})\|^2$$

$$\leq -\beta \|\nabla F_{t,\eta}(x_t, y_t^*(x_t))\|^2 - \beta \langle \nabla F_{t,\eta}(x_t, y_t^*(x_t)), \widehat{\nabla} F_{t,\eta}(x_t, y_{t+1}) - \nabla F_{t,\eta}(x_t, y_t^*(x_t)) \rangle$$

$$+ \frac{\beta^2 L_f}{2} \|\widehat{\nabla} F_{t,\eta}(x_t, y_{t+1})\|^2$$

$$\leq -\left( \frac{\beta}{2} - \beta^2 L_f \right) \|\nabla F_{t,\eta}(x_t, y_t^*(x_t))\|^2$$

$$+ \left( \frac{\beta}{2} + \beta^2 L_f \right) \|\nabla F_{t,\eta}(x_t, y_t^*(x_t)) - \widehat{\nabla} F_{t,\eta}(x_t, y_{t+1})\|^2$$

such that

$$\left( \frac{\beta}{2} - \beta^2 L_f \right) \sum_{t=1}^{T} \|\nabla F_{t,\eta}(x_t, y_t^*(x_t))\|^2 \leq \underbrace{\sum_{t=1}^{T} [F_{t,\eta}(x_t, y_t^*(x_t)) - F_{t,\eta}(x_{t+1}, y_t^*(x_{t+1}))]}_{(a)}$$

$$+ \left( \frac{\beta}{2} + \beta^2 L_f \right) \underbrace{\sum_{t=1}^{T} \|\nabla F_{t,\eta}(x_t, y_t^*(x_t)) - \widehat{\nabla} F_{t,\eta}(x_t, y_{t+1})\|^2}_{(b)}. \tag{11}$$

(1) We first have the following lemma to bound the term (a) from above:

**Lemma H.3.** *The following inequality holds:*

$$\sum_{t=1}^{T} [F_{t,\eta}(x_t, y_t^*(x_t)) - F_{t,\eta}(x_{t+1}, y_t^*(x_{t+1}))] \leq \frac{2MT}{W} + V_{1,T}.$$

*Proof.* First, it is clear that

$$F_{t,\eta}(x_t, y_t^*(x_t)) - F_{t,\eta}(x_{t+1}, y_t^*(x_{t+1}))$$

$$= \frac{1}{W} \sum_{i=0}^{K-1} \eta^i f_{t-i}(x_{t-i}, y_{t-i}^*(x_{t-i})) - \frac{1}{W} \sum_{i=0}^{K-1} \eta^i f_{t-i}(x_{t+1-i}, y_{t-i}^*(x_{t+1-i}))$$

$$= \underbrace{\frac{1}{W} \sum_{i=0}^{K-1} \eta^i [f_{t-i}(x_{t-i}, y_{t-i}^*(x_{t-i})) - f_{t+1-i}(x_{t+1-i}, y_{t+1-i}^*(x_{t+1-i}))]}_{(a.1)}$$

$$+ \frac{1}{W} \sum_{i=0}^{K-1} \eta^i [f_{t+1-i}(x_{t+1-i}, y^*_{t+1-i}(x_{t+1-i})) - f_{t-i}(x_{t+1-i}, y^*_{t-i}(x_{t+1-i}))]. \quad (12)$$

$$\underbrace{\phantom{\frac{1}{W} \sum_{i=0}^{K-1} \eta^i [f_{t+1-i}(x_{t+1-i}, y^*_{t+1-i}(x_{t+1-i})) - f_{t-i}(x_{t+1-i}, y^*_{t-i}(x_{t+1-i}))]}}_{(a.2)}$$

For the term (a.1), we can obtain that

$$\frac{1}{W} \sum_{i=0}^{K-1} \eta^i [f_{t-i}(x_{t-i}, y^*_{t-i}(x_{t-i})) - f_{t+1-i}(x_{t+1-i}, y^*_{t+1-i}(x_{t+1-i}))]$$

$$= \frac{1}{W} [f_t(x_t, y^*_t(x_t)) + \eta f_{t-1}(x_{t-1}, y^*_{t-1}(x_{t-1})) + ... + \eta^{K-1} f_{t+1-K}(x_{t+1-K}, y^*_{t+1-K}(x_{t+1-K}))$$

$$- f_{t+1}(x_{t+1}, y^*_{t+1}(x_{t+1}) - \eta f_t(x_t, y^*_t(x_t)) - ... - \eta^{K-1} f_{t+2-K}(x_{t+2-K}, y^*_{t+2-K}(x_{t+2-K})))]$$

$$= \frac{1}{W} [\eta^{K-1} f_{t+1-K}(x_{t+1-K}, y^*_{t+1-K}(x_{t+1-K})) - f_{t+1}(x_{t+1}, y^*_{t+1}(x_{t+1}))]$$

$$+ \frac{1}{W} \sum_{i=1}^{K-1} (\eta^{i-1} - \eta^i) f_{t+1-i}(x_{t+1-i}, y^*_{t+1-i}(x_{t+1-i}))$$

$$\leq \frac{M(1 + \eta^{K-1})}{W} + \frac{M}{W} \sum_{i=1}^{K-1} (\eta^{i-1} - \eta^i)$$

$$= \frac{2M}{W} \quad (13)$$

where the inequality holds because of Assumption 5.3.

For the term (a.2), we can have that

$$\frac{1}{W} \sum_{i=0}^{K-1} \eta^i [f_{t+1-i}(x_{t+1-i}, y^*_{t+1-i}(x_{t+1-i})) - f_{t-i}(x_{t+1-i}, y^*_{t-i}(x_{t+1-i}))]$$

$$\leq \frac{1}{W} \sum_{i=0}^{K-1} \eta^i \sup_x [f_{t+1-i}(x, y^*_{t+1-i}(x)) - f_{t-i}(x, y^*_{t-i}(x))]. \quad (14)$$

By substituting Equation (13) and Equation (14) back to Equation (12), Lemma H.3 can be proved:

$$\sum_{t=1}^{T} [F_{t,\eta}(x_t, y^*_t(x_t)) - F_{t,\eta}(x_{t+1}, y^*_t(x_{t+1}))]$$

$$\leq \sum_{t=1}^{T} \left[ \frac{2M}{W} + \frac{1}{W} \sum_{i=0}^{K-1} \eta^i \sup_x [f_{t+1-i}(x, y^*_{t+1-i}(x)) - f_{t-i}(x, y^*_{t-i}(x))] \right]$$

$$\leq \frac{2MT}{W} + V_{1,T}.$$

$\square$

(2) Next, for the term (b) which captures the window-averaged hypergradient estimation error, it follows that

$$\sum_{t=1}^{T} \|\nabla F_{t,\eta}(x_t, y^*_t(x_t)) - \widehat{\nabla} F_{t,\eta}(x_t, y_{t+1})\|^2$$

$$= \sum_{t=1}^{T} \left\| \frac{1}{W} \sum_{i=0}^{K-1} \eta^i [\nabla f_{t-i}(x_{t-i}, y^*_{t-i}(x_{t-i})) - \widehat{\nabla} f_{t-i}(x_{t-i}, y_{t+1-i})] \right\|^2$$

$$= \sum_{t=1}^{T} \Bigg[ \sum_{i=0}^{K-1} \frac{\eta^i}{W} \sum_{j=0}^{K-1} \frac{\eta^j}{W} \langle \nabla f_{t-i}(x_{t-i}, y_{t-i}^*(x_{t-i})) - \widehat{\nabla} f_{t-i}(x_{t-i}, y_{t+1-i}),$$

$$\nabla f_{t-j}(x_{t-j}, y_{t-j}^*(x_{t-j})) - \widehat{\nabla} f_{t-j}(x_{t-j}, y_{t+1-j}) \rangle \Bigg]$$

$$\leq \sum_{t=1}^{T} \Bigg[ \sum_{i=0}^{K-1} \frac{\eta^i}{W} \sum_{j=0}^{K-1} \frac{\eta^j}{W} \Big[ \frac{1}{2} \left\| \nabla f_{t-i}(x_{t-i}, y_{t-i}^*(x_{t-i})) - \widehat{\nabla} f_{t-i}(x_{t-i}, y_{t+1-i}) \right\|^2$$

$$+ \frac{1}{2} \left\| \nabla f_{t-j}(x_{t-j}, y_{t-j}^*(x_{t-j})) - \widehat{\nabla} f_{t-j}(x_{t-j}, y_{t+1-j}) \right\|^2 \Big] \Bigg]$$

$$= \sum_{t=1}^{T} \Bigg[ \frac{1}{W} \sum_{i=0}^{K-1} \eta^i \left\| \nabla f_{t-i}(x_{t-i}, y_{t-i}^*(x_{t-i})) - \widehat{\nabla} f_{t-i}(x_{t-i}, y_{t+1-i}) \right\|^2 \Bigg], \tag{15}$$

which boils down to characterize the hypergradient estimation error $\| \nabla f_t(x_t, y_t^*(x_t)) - \widehat{\nabla} f_t(x_t, y_{t+1}) \|^2$ on the outer level objective function at each round.

By leveraging Theorem 5.6, it is clear that

$$\sum_{t=2}^{T} \| \nabla F_{t,\eta}(x_t, y_t^*(x_t)) - \widehat{\nabla} F_{t,\eta}(x_t, y_{t+1}) \|^2$$

$$\leq \sum_{t=2}^{T} \Bigg[ \frac{1}{W} \sum_{i=0}^{K-1} \eta^i \| \nabla f_{t-i}(x_{t-i}, y_{t-i}^*(x_{t-i})) - \widehat{\nabla} f_{t-i}(x_{t-i}, y_{t+1-i}) \|^2 \Bigg]$$

$$= \sum_{t=2}^{T} \Bigg[ \frac{3L_1^2}{W} \sum_{i=0}^{K-1} \eta^i \delta_{t-i} \Bigg]$$

$$\leq \frac{3L_1^2}{W} \sum_{t=2}^{T} \sum_{i=0}^{K-1} \eta^i \Bigg\{ \underbrace{\left(1 - \frac{\alpha \mu_g}{2}\right)^{t-1-i} \delta_1}_{(b.1)}$$

$$+ \underbrace{G_2 \sum_{j=0}^{t-i-2} \left(1 - \frac{\alpha \mu_g}{2}\right)^j \| y_{t-1-i-j}^*(x_{t-1-i-j}) - y_{t-i-j}^*(x_{t-1-i-j}) \|^2}_{(b.2)}$$

$$+ \underbrace{\frac{2L_1^2 G_2}{\mu_g^2} \sum_{j=0}^{t-2-i} \left(1 - \frac{\alpha \mu_g}{2}\right)^j \| x_{t-1-i-j} - x_{t-i-j} \|^2}_{(b.3)} \Bigg\}. \tag{16}$$

Let $\gamma = \frac{\alpha}{2}$. For the first term (b.1), it can be seen that for $1 - \frac{\alpha \mu_g}{2} < \eta < 1$

$$\sum_{t=2}^{T} \sum_{i=0}^{K-1} \eta^i (1 - \gamma \mu_g)^{t-1-i} \delta_1 = \delta_1 \sum_{t=2}^{T} \sum_{i=0}^{K-1} \left[ \left( \frac{1 - \gamma \mu_g}{\eta} \right)^{t-1-i} \eta^{t-1} \right]$$

$$\leq \delta_1 \sum_{t=2}^{T} \eta^{t-1} \frac{\frac{1 - \gamma \mu_g}{\eta}}{1 - \frac{1 - \gamma \mu_g}{\eta}}$$

$$= \frac{\delta_1 (1 - \gamma \mu_g)}{\eta - 1 + \gamma \mu_g} \frac{\eta (1 - \eta^{T-1})}{1 - \eta}$$

$$\leq \frac{\delta_1 \eta (1 - \gamma \mu_g)}{(1 - \eta)(\eta - 1 + \gamma \mu_g)}. \tag{17}$$

For the second term (b.2), we have

$$\sum_{i=0}^{K-1} \eta^i \sum_{j=0}^{t-2-i} (1-\gamma\mu_g)^j \|y_{t-1-i-j}^*(x_{t-1-i-j}) - y_{t-i-j}^*(x_{t-1-i-j})\|^2$$

$$= \sum_{j=0}^{t-2} (1-\gamma\mu_g)^j \|y_{t-1-j}^*(x_{t-1-j}) - y_{t-j}^*(x_{t-1-j})\|^2$$

$$+ \sum_{j=0}^{t-3} \eta(1-\gamma\mu_g)^j \|y_{t-2-j}^*(x_{t-2-j}) - y_{t-1-j}^*(x_{t-2-j})\|^2$$

$$+ \cdots$$

$$+ \sum_{j=0}^{t-1-K} \eta^{K-1}(1-\gamma\mu_g)^j \|y_{t-K-j}^*(x_{t-K-j}) - y_{t+1-K-j}^*(x_{t-K-j})\|^2$$

$$= \|y_{t-1}^*(x_{t-1}) - y_t^*(x_{t-1})\|^2 + [(1-\gamma\mu_g)+\eta]\|y_{t-2}^*(x_{t-2}) - y_{t-1}^*(x_{t-2})\|^2$$

$$+ [(1-\gamma\mu_g)^2 + \eta(1-\gamma\mu_g) + \eta^2]\|y_{t-3}^*(x_{t-3}) - y_{t-2}^*(x_{t-3})\|^2$$

$$+ \cdots$$

$$+ [(1-\gamma\mu_g)^{K-2} + (1-\gamma\mu_g)^{K-3}\eta + \ldots + \eta^{K-2}]\|y_{t+1-K}^*(x_{t+1-K}) - y_{t+2-K}^*(x_{t+1-K})\|^2$$

$$+ \sum_{j=0}^{t-1-K} \left\{ [(1-\gamma\mu_g)^{j+K-1} + \eta(1-\gamma\mu_g)^{j+K-2} + \ldots + \eta^{K-1}(1-\gamma\mu_g)^j] \right.$$

$$\left. \cdot \|y_{t-K-j}^*(x_{t-K-j}) - y_{t+1-K-j}^*(x_{t-K-j})\|^2 \right\}$$

$$= \|y_{t-1}^*(x_{t-1}) - y_t^*(x_{t-1})\|^2 + \frac{1-(\frac{1-\gamma\mu_g}{\eta})^2}{1-\frac{1-\gamma\mu_g}{\eta}}\eta\|y_{t-2}^*(x_{t-2}) - y_{t-1}^*(x_{t-2})\|^2$$

$$+ \frac{1-(\frac{1-\gamma\mu_g}{\eta})^3}{1-\frac{1-\gamma\mu_g}{\eta}}\eta^2\|y_{t-3}^*(x_{t-3}) - y_{t-2}^*(x_{t-3})\|^2 + \cdots$$

$$+ \frac{1-(\frac{1-\gamma\mu_g}{\eta})^{K-1}}{1-\frac{1-\gamma\mu_g}{\eta}}\eta^{K-2}\|y_{t+1-K}^*(x_{t+1-K}) - y_{t+2-K}^*(x_{t+1-K})\|^2$$

$$+ \frac{1-(\frac{1-\gamma\mu_g}{\eta})^K}{1-\frac{1-\gamma\mu_g}{\eta}}\eta^{K-1} \sum_{j=0}^{t-1-K} (1-\gamma\mu_g)^j \|y_{t-K-j}^*(x_{t-K-j}) - y_{t+1-K-j}^*(x_{t-K-j})\|^2$$

$$\leq \frac{1}{1-\frac{1-\gamma\mu_g}{\eta}}\left[ \|y_{t-1}^*(x_{t-1}) - y_t^*(x_{t-1})\|^2 + \eta\|y_{t-2}^*(x_{t-2}) - y_{t-1}^*(x_{t-2})\|^2 \right.$$

$$+ \cdots + \eta^{K-2}\|y_{t+1-K}^*(x_{t+1-K}) - y_{t+2-K}^*(x_{t+1-K})\|^2$$

$$\left. + \sum_{j=0}^{t-1-K} \eta^{K-1+j}\|y_{t-K-j}^*(x_{t-K-j}) - y_{t+1-K-j}^*(x_{t-K-j})\|^2 \right],$$

such that

$$\sum_{t=2}^{T}\sum_{i=0}^{K-1} \eta^i \sum_{j=0}^{t-2-i} (1-\gamma\mu_g)^j \|y_{t-1-i-j}^*(x_{t-1-i-j}) - y_{t-i-j}^*(x_{t-1-i-j})\|^2$$

$$\leq \frac{\eta}{\eta-1+\gamma\mu_g} \sum_{t=2}^{T}\left[ \|y_{t-1}^*(x_{t-1}) - y_t^*(x_{t-1})\|^2 + \eta\|y_{t-2}^*(x_{t-2}) - y_{t-1}^*(x_{t-2})\|^2 \right.$$

$$+ \cdots + \eta^{K-2} \|y^*_{t+1-K}(x_{t+1-K}) - y^*_{t+2-K}(x_{t+1-K})\|^2$$

$$+ \sum_{j=0}^{t-1-K} \eta^{K-1+j} \|y^*_{t-K-j}(x_{t-K-j}) - y^*_{t+1-K-j}(x_{t-K-j})\|^2 \Bigg]$$

$$\leq \frac{\eta}{(1-\eta)(\eta - 1 + \gamma\mu_g)} \sum_{t=2}^{T} \sup_{x} \|y^*_{t-1}(x) - y^*_t(x)\|^2$$

$$= \frac{\eta}{(1-\eta)(\eta - 1 + \gamma\mu_g)} H_{2,T} \tag{18}$$

where $H_{2,T} = \sum_{t=2}^{T} \sup_x \|y^*_{t-1}(x) - y^*_t(x)\|^2$.

Besides, we know that

$$\|x_{t-1} - x_t\|^2$$

$$= \beta^2 \|\widehat{\nabla} F_{t-1}(x_{t-1}, y_t)\|^2$$

$$\leq 2\beta^2 \|\nabla F_{t-1}(x_{t-1}, y^*_{t-1}(x_{t-1})) - \widehat{\nabla} F_{t-1}(x_{t-1}, y_t)\|^2 + 2\beta^2 \|\nabla F_{t-1}(x_{t-1}, y^*_{t-1}(x_{t-1}))\|^2.$$

Following the same analysis for the second term (b.2), the following result can be obtained that for the third term (b.3):

$$\sum_{t=2}^{T} \sum_{i=0}^{K-1} \eta^i \sum_{j=0}^{t-2-i} (1 - \gamma\mu_g)^j \|x_{t-1-i-j} - x_{t-i-j}\|^2$$

$$\leq \frac{\eta}{\eta - 1 + \gamma\mu_g} \sum_{t=2}^{T} \Bigg[ \|x_{t-1} - x_t\|^2 + \eta \|x_{t-2} - x_{t-1}\|^2$$

$$+ \cdots + \eta^{K-2} \|x_{t+1-K} - x_{t+2-K}\|^2$$

$$+ \sum_{j=0}^{t-1-K} \eta^{K-1+j} \|x_{t-K-j} - x_{t+1-K-j}\|^2 \Bigg]$$

$$\leq \frac{2\beta^2 \eta}{\eta - 1 + \gamma\mu_g} \sum_{t=2}^{T} \Bigg[ \|\nabla F_{t-1}(x_{t-1}, y^*_{t-1}(x_{t-1})) - \widehat{\nabla} F_{t-1}(x_{t-1}, y_t)\|^2$$

$$+ \|\nabla F_{t-1}(x_{t-1}, y^*_{t-1}(x_{t-1}))\|^2$$

$$+ \eta \Big( \|\nabla F_{t-2}(x_{t-2}, y^*_{t-2}(x_{t-2})) - \widehat{\nabla} F_{t-2}(x_{t-2}, y_{t-1})\|^2 + \|\nabla F_{t-2}(x_{t-2}, y^*_{t-2}(x_{t-2}))\|^2 \Big)$$

$$+ \cdots$$

$$+ \eta^{K-2} \Big( \|\nabla F_{t+1-K}(x_{t+1-K}, y^*_{t+1-K}(x_{t+1-K})) - \widehat{\nabla} F_{t+1-K}(x_{t+1-K}, y_{t+2-K})\|^2$$

$$+ \|\nabla F_{t+1-K}(x_{t+1-K}, y^*_{t+1-K}(x_{t+1-K}))\|^2 \Big)$$

$$+ \sum_{j=0}^{t-1-K} \eta^{K-1+j} \Big( \|\nabla F_{t-K-j}(x_{t-K-j}, y^*_{t-K-j}(x_{t-K-j})) - \widehat{\nabla} F_{t-K-j}(x_{t-K-j}, y_{t+1-K-j})\|^2$$

$$+ \|\nabla F_{t-K-j}(x_{t-K-j}, y^*_{t-K-j}(x_{t-K-j}))\|^2 \Big) \Bigg]$$

$$\leq \frac{2\beta^2 \eta}{\eta - 1 + \gamma\mu_g} \frac{1 - \eta^{T-1}}{1 - \eta} \sum_{t=2}^{T} \|\nabla F_{t-1}(x_{t-1}, y^*_{t-1}(x_{t-1})) - \widehat{\nabla} F_{t-1}(x_{t-1}, y_t)\|^2$$

$$+ \frac{2\beta^2 \eta}{\eta - 1 + \gamma\mu_g} \frac{1 - \eta^{T-1}}{1 - \eta} \sum_{t=2}^{T} \|\nabla F_{t-1}(x_{t-1}, y^*_{t-1}(x_{t-1}))\|^2$$

$$\leq \frac{2\beta^2\eta}{(1-\eta)(\eta-1+\gamma\mu_g)} \sum_{t=2}^{T} \|\nabla F_{t-1}(x_{t-1}, y_{t-1}^*(x_{t-1})) - \widehat{\nabla} F_{t-1}(x_{t-1}, y_t)\|^2$$

$$+ \frac{2\beta^2\eta}{(1-\eta)(\eta-1+\gamma\mu_g)} \sum_{t=2}^{T} \|\nabla F_{t-1}(x_{t-1}, y_{t-1}^*(x_{t-1}))\|^2. \tag{19}$$

Therefore, by substituting Equation (17), Equation (18) and Equation (19) into Equation (16), we can have that

$$\sum_{t=1}^{T} \|\nabla F_{t,\eta}(x_t, y_t^*(x_t)) - \widehat{\nabla} F_{t,\eta}(x_t, y_{t+1})\|^2$$

$$\leq \frac{3L_1^2}{W} \sum_{t=2}^{T} \sum_{i=0}^{K-1} \eta^i \Bigg\{ (1-\gamma\mu_g)^{t-1-i}\delta_1$$

$$+ G_2 \sum_{j=0}^{t-i-2} (1-\gamma\mu_g)^j \|y_{t-1-i-j}^*(x_{t-1-i-j}) - y_{t-i-j}^*(x_{t-1-i-j})\|^2$$

$$+ \frac{2L_1^2 G_2}{\mu_g^2} \sum_{j=0}^{t-2-i} (1-\gamma\mu_g)^j \|x_{t-1-i-j} - x_{t-i-j}\|^2 \Bigg\} + \|\nabla f_1(x_1, y_1^*(x_1)) - \widehat{\nabla} f_1(x_1, y_2)\|^2$$

$$\leq \frac{3L_1^2}{W} \Bigg[ \frac{\delta_1\eta(1-\gamma\mu_g)}{(1-\eta)(\eta-1+\gamma\mu_g)} + G_2 \frac{\eta}{(1-\eta)(\eta-1+\gamma\mu_g)} H_{2,T}$$

$$+ \frac{2L_1^2 G_2}{\mu_g^2} \frac{2\beta^2\eta}{(1-\eta)(\eta-1+\gamma\mu_g)} \sum_{t=1}^{T} \|\nabla F_t(x_t, y_t^*(x_t)) - \widehat{\nabla} F_t(x_t, y_{t+1})\|^2$$

$$+ \frac{2L_1^2 G_2}{\mu_g^2} \frac{2\beta^2\eta}{(1-\eta)(\eta-1+\gamma\mu_g)} \sum_{t=1}^{T} \|\nabla F_t(x_t, y_t^*(x_t))\|^2 \Bigg] + \|\nabla f_1(x_1, y_1^*(x_1)) - \widehat{\nabla} f_1(x_1, y_2)\|^2$$

which gives that

$$\left( 1 - \frac{12L_1^4\beta^2 G_2\eta}{\mu_g^2 W(1-\eta)(\eta-1+\gamma\mu_g)} \right) \sum_{t=1}^{T} \|\nabla F_{t,\eta}(x_t, y_t^*(x_t)) - \widehat{\nabla} F_{t,\eta}(x_t, y_{t+1})\|^2$$

$$\leq \frac{3L_1^2\eta}{W(1-\eta)(\eta-1+\gamma\mu_g)} \Bigg[ \delta_1(1-\gamma\mu_g) + G_2 H_{2,T} + \frac{4L_1^2 G_2\beta^2}{\mu_g^2} \sum_{t=1}^{T} \|\nabla F_t(x_t, y_t^*(x_t))\|^2 \Bigg]$$

$$+ \|\nabla f_1(x_1, y_1^*(x_1)) - \widehat{\nabla} f_1(x_1, y_2)\|^2.$$

Here $1 - \frac{12L_1^4\beta^2 G_2\eta}{\mu_g^2 W(1-\eta)(\eta-1+\gamma\mu_g)} \geq \frac{1}{2}$ because

$$\beta^2 \leq \frac{\beta}{L_f} \leq \frac{\mu_g^2 L_f W(1-\eta)(\eta-1-\alpha\mu_g/2)}{24L_1^4 G_2\eta} L_f$$

$$= \frac{\mu_g^2 W(1-\eta)(\eta-1-\alpha\mu_g/2)}{24L_1^4 G_2\eta}.$$

Let $G_3 = 1 - \frac{12L_1^4\beta^2 G_2\eta}{\mu_g^2 W(1-\eta)(\eta-1+\gamma\mu_g)}$ and $G_4 = \frac{3L_1^2\eta}{W(1-\eta)(\eta-1+\gamma\mu_g)}$. It is clear that

$$\sum_{t=1}^{T} \|\nabla F_{t,\eta}(x_t, y_t^*(x_t)) - \widehat{\nabla} F_{t,\eta}(x_t, y_{t+1})\|^2$$

$$\leq \frac{G_4}{G_3} [\delta_1(1-\gamma\mu_g) + G_2 H_{2,T}] + \frac{G_4}{G_3} \frac{4L_1^2 G_2\beta^2}{\mu_g^2} \sum_{t=1}^{T} \|\nabla F_t(x_t, y_t^*(x_t))\|^2$$

$$+ \frac{1}{G_3} \|\nabla f_1(x_1, y_1^*(x_1)) - \widehat{\nabla} f_1(x_1, y_2)\|^2.$$

Based on Equation (11), we can have

$$\left(\frac{\beta}{2} - \beta^2 L_f\right) \sum_{t=1}^{T} \|\nabla F_{t,\eta}(x_t, y_t^*(x_t))\|^2$$

$$\leq \sum_{t=1}^{T} [F_{t,\eta}(x_t, y_t^*(x_t)) - F_{t,\eta}(x_{t+1}, y_t^*(x_{t+1}))]$$

$$+ \left(\frac{\beta}{2} + \beta^2 L_f\right) \sum_{t=1}^{T} \|\nabla F_{t,\eta}(x_t, y_t^*(x_t)) - \widehat{\nabla} F_{t,\eta}(x_t, y_{t+1})\|^2$$

$$\leq \sum_{t=1}^{T} [F_{t,\eta}(x_t, y_t^*(x_t)) - F_{t,\eta}(x_{t+1}, y_t^*(x_{t+1}))] + \left(\frac{\beta}{2} + \beta^2 L_f\right) \left[ \frac{G_4}{G_3}[\delta_1(1 - \gamma\mu_g) + G_2 H_{2,T}] \right.$$

$$\left. + \frac{G_4}{G_3} \frac{4L_1^2 G_2 \beta^2}{\mu_g^2} \sum_{t=1}^{T} \|\nabla F_t(x_t, y_t^*(x_t))\|^2 + \frac{1}{G_3} \|\nabla f_1(x_1, y_1^*(x_1)) - \widehat{\nabla} f_1(x_1, y_2)\|^2 \right]$$

such that

$$\sum_{t=1}^{T} \|\nabla F_{t,\eta}(x_t, y_t^*(x_t))\|^2$$

$$\leq \frac{1}{\frac{\beta}{2} - \beta^2 L_f - \frac{G_4}{G_3} \frac{4L_1^2 G_2 \beta^2}{\mu_g^2} \left(\frac{\beta}{2} + \beta^2 L_f\right)} \sum_{t=1}^{T} [F_{t,\eta}(x_t, y_t^*(x_t)) - F_{t,\eta}(x_{t+1}, y_t^*(x_{t+1}))]$$

$$+ \frac{\left(\frac{1}{2} + \beta L_f\right) \left\{ \frac{G_4}{G_3}[\delta_1(1 - \gamma\mu_g) + G_2 H_{2,T}] + \frac{1}{G_3} \|\nabla f_1(x_1, y_1^*(x_1)) - \widehat{\nabla} f_1(x_1, y_2)\|^2 \right\}}{\frac{1}{2} - \beta L_f - \frac{G_4}{G_3} \frac{4L_1^2 G_2 \beta}{\mu_g^2} \left(\frac{\beta}{2} + \beta^2 L_f\right)}$$

$$\leq \frac{\frac{2MT}{W} + V_{1,T}}{\frac{\beta}{2} - \beta^2 L_f - \frac{G_4}{G_3} \frac{4L_1^2 G_2 \beta^2}{\mu_g^2} \left(\frac{\beta}{2} + \beta^2 L_f\right)} + \frac{\left(\frac{1}{2} + \beta L_f\right) \frac{G_4 G_2 H_{2,T}}{G_3}}{\frac{1}{2} - \beta L_f - \frac{G_4}{G_3} \frac{4L_1^2 G_2 \beta}{\mu_g^2} \left(\frac{\beta}{2} + \beta^2 L_f\right)}$$

$$+ \frac{\left(\frac{1}{2} + \beta L_f\right) \left\{ \frac{G_4}{G_3} \delta_1(1 - \gamma\mu_g) + \frac{1}{G_3} \|\nabla f_1(x_1, y_1^*(x_1)) - \widehat{\nabla} f_1(x_1, y_2)\|^2 \right\}}{\frac{1}{2} - \beta L_f - \frac{G_4}{G_3} \frac{4L_1^2 G_2 \beta}{\mu_g^2} \left(\frac{\beta}{2} + \beta^2 L_f\right)}$$

$$= O\left( \frac{T}{\beta W} + \frac{V_{1,T}}{\beta} + H_{2,T} \right).$$

Here $\frac{1}{2} - \beta L_f - \frac{G_4}{G_3} \frac{4L_1^2 G_2 \beta}{\mu_g^2} \left(\frac{\beta}{2} + \beta^2 L_f\right) > 0$ because

$$\frac{1}{2} - \beta L_f - \frac{G_4}{G_3} \frac{4L_1^2 G_2 \beta}{\mu_g^2} \left(\frac{\beta}{2} + \beta^2 L_f\right)$$

$$\overset{(a)}{>} \frac{1}{2} - \beta L_f - \frac{G_4}{G_3} \frac{4L_1^2 G_2 \beta^2}{\mu_g^2}$$

$$\overset{(b)}{\geq} \frac{1}{2} - 2\beta L_f$$

$$\overset{(c)}{\geq} 0,$$

where (a) and (c) are because $\beta L_f < \frac{1}{4}$, and (b) is because $\beta L_f > \frac{G_4}{G_3} \frac{4L_1^2 G_2 \beta^2}{\mu_g^2}$.

