# OpenReview forum: "Non-Convex Bilevel Optimization with Time-Varying Objective Functions"
_NeurIPS.cc/2023/Conference — NeurIPS 2023 poster_

### Official Review · Reviewer_ouS8 · 2023-07-06

**Soundness:** 3 good
**Presentation:** 4 excellent
**Contribution:** 3 good
**Rating:** 7
**Confidence:** 3

**Summary:**

This paper studies bilevel optimization in an online setting where the objective in both the levels are allowed to vary with time, and the goal is to develop an algorithm with sublinear regret. This paper proposes a practical single-loop algorithm that updates the lower-level variable only once for each upper-level variable update. The lower-level variable is updated via vanilla SGD, while the hypergradient for the upper-level update is computed via some specified number of steps of Conjugate Gradient, and averaged over a window of specified size. This design significantly improves the computational and memory overhead relative to the existing baseline which requires access to all the past objectives and gradient oracles in the window; the proposed scheme just needs to maintain the hypergradients in the window. Based a novel notion of bilevel regret, the paper shows that the proposed algorithm is able to achieve sublinear regret under some standard assumptions for appropriately set algorithmic parameters and window sizes. The empirical evaluation on two online bilevel applications show that the proposed scheme can match the performance of the existing baseline while being significantly more efficient both in terms of time and memory requirement -- the computational advantage of the proposed scheme is enhanced when considering larger window sizes.

**Strengths:**

**Practical online solution.**
In the online bilevel setting, it seems more practical to have a single-loop algorithm since the objectives/gradients for each of the level will probably be made available to the learner in a sequential alternate manner. However, single loop bilevel algorithms are usually harder to analyse. So it is a significant contribution to have a single loop algorithm that has sublinear regret.

**Intuitive presentation of theoretical analyses.**
The authors have presented the theoretical analyses in a very intuitive and clear manner. After specifying the necessary assumptions, the authors discuss the necessary steps to complete the analyses, and gradually build up to each of the results. This presentation is very easy to follow for a reader, and I really appreciate the work done for such a presentation. After the main theorem, the author clearly discuss the conditions under which we might achieve the desired sublinear regret.

**Strong empirical performance against baseline.**
The experimental results show clearly show that the proposed SOBOW matches the regret of the OAGD baseline, but is able to do so with significantly lower computational and memory overhead, and without access to the past objectives. The computational gains are very significant, with up to almost $20\times$ speedup. This is an impressive result, making the solution even more practically useful.

**Weaknesses:**

**Hyperparameters in the definition of regret.**
One of the weaknesses of this paper is that the proposed novel notion of bilevel regret itself (equation (2)) seems to depend on the window size $K$ and the decay rate $\eta$ (and it is not quite clear what the subscript $w$ in $BLR_w(T)$ denotes). Given that the subsequent analysis shows that these quantities need to be set appropriately for the desired convergence rate, it is odd that the notion of regret itself depends on it. The algorithm can use such hyperparameters, but the term quantifying the regret should not depend on them. Is this standard in dynamic local regret analysis? One would expect that the bilevel local regret would be defined as some term such as $\sum_{t=1}^T || \nabla_x f_t(x_t, y_t^*(x_t)) ||^2$ or something similar, where we are computing the per-time-step (local) suboptimality, and we would want this quantity to grow sublinearly with $T$ (with appropriate assumptions regarding the relationship between $f_t, g_t$ and $f_{t+1}, g_{t+1}$). It seems as if such a definition of bilevel regret (equation (2)) was considered because it seems to match the form of the upper-level update that is used in the proposed algorithm.

**No dependence on lower-level suboptimality in the regret.**
Another issue with the considered notion of bilevel regret is that it is not clear why it is meaningful for the regret to depend on $(x_t, y_t^*(x_t))$ instead of just $(x_t, y_t)$ or $(x_t, y_{t+1})$. Alternately, it is not clear why the sub-optimality in the lower-level decision variable (that is, having $y_{t+1}$ instead of $y_t^*(x_t)$) does not contribute to the regret in any way. Analyses of static bilevel optimization usually establish convergence of $ || \nabla_x f(x_t, y_t^*(x_t)) ||^2$ **as well as** that of $|| y_{t+1} - y_t^*(x_t) ||^2$.

**Questions:**

- Given existing single-loop static algorithms such as TTSA [A] and STABLE [B], how is the proposed algorithm positioned against these? One of the challenges with single-loop schemes is that the upper-level updates need to be very slow (that is, have a small upper-level learning rate relative to the lower-level learning rate) if we are just using a single SGD step lower-level update. This is because, otherwise, it is hard to guarantee that $y_{t+1}$ converges to $y^*(x_t)$ since the $x_{t-1} \to x_t$ update can significantly move the lower-level target from $y^*(x_{t-1}) \to y^*(x_t)$, which a single SGD step is unable to catch up with. That is why, the more expensive but sophisticated lower-level update is proposed in STABLE, to allow for faster upper-level updates. Does this issue manifest in the proposed SOBOW, resulting in a need for a smaller upper-level learning rate (and thus slower convergence), or is there something in the nature of the online bilevel setup that mitigates this issue?
- Given that $\{Q_t, t \in [T]\}$ would be an increasing sequence, what is the motivation to not just solve the least-squares problem to sufficient optimality at each step and remove the error term in Lemma 5.5, and simplify the analysis?

>[A] Hong, Mingyi, et al. "A two-timescale stochastic algorithm framework for bilevel optimization: Complexity analysis and application to actor-critic." SIAM Journal on Optimization 33.1 (2023): 147-180.

>[B] Chen, Tianyi, et al. "A single-timescale method for stochastic bilevel optimization." International Conference on Artificial Intelligence and Statistics. PMLR, 2022.

**Limitations:**

The authors do explicitly discuss some limitations, and I do not anticipate any potential negative societal impact of this work.

---

> ### Author Rebuttal · Authors · 2023-08-09
>
> Thank you for your thorough reviews and constructive comments. We provide our response to your comments below. If our response resolves your concern, we would greatly appreciate it if you could consider increasing your score.
>
> Q1: Hyperparameters in the definition of regret. Is it standard in dynamic regret analysis?
>
> A1: Many thanks for the insightful comments. The window-smoothed regret definition is widely adopted in online nonconvex optimization for dynamic local regret analysis (e.g., E. Hazan et al., 2017; S. Aydore et al., 2019; N. Hallak et al., 2021; D. Tarzanagh and L. Balzano, 2022; Y. Huang et al., 2023; Z. Guan et al., 2023), and our definition follows exactly the same way as in S. Aydore et al., 2019 and D. Tarzanagh and L. Balzano, 2022, which also have both window size and $\eta$ in their definitions. The underlying ideas about designing such a local regret are as follows:
>
> (1) It has been shown in the literature of online learning (e.g, E. Hazan et al., 2017) that this time-smoothing is indeed necessary for the definition of the local regret in the non-convex setup. Specifically, for any online algorithm, there exists an adversarial sequence of loss functions,  which can force the local regret to be $\Omega(\frac{T}{K^2})$. Therefore, a sublinear regret cannot be achieved for $\sum_{t=1}^T ||\nabla_x f_t(x_t, y_t^*(x_t))||^2$ suggested by the reviewer (which corresponds to the case $K=1$). In practice, the average performance of a system is also a typical and intuitive notion that is commonly used to evaluate real-world applications. For example, under changing environments, such an average performance metric during a period is naturally adopted in time series forecasting problems (S. Aydore et al., 2019). In terms of the decaying rate $\eta$, it is reasonable to assign larger weights to the most recent functions, in the same way with, e.g., S. Aydore et al., 2019 and D. Tarzanagh and L. Balzano, 2022. Here the subscript $w$ in $BLR_w(T)$ just refers to the window-averaged local regret.
>
> (2) A small time-smoothed gradient in expectation implies that the outer-level decision is becoming better and closer to the local optima for the outer-level optimization problem at each round.
>
> - E. Hazan et al.. Efficient regret minimization in non-convex games. ICML, 2017.
> - S. Aydore et al.. Dynamic local regret for non-convex online forecasting. NeurIPS, 2019.
> - N. Hallak et al.. Regret minimization in stochastic non-convex learning via a proximal-gradient approach. ICML, 2021.
> - D. Tarzanagh and L. Balzano. Online bilevel optimization: regret analysis of online alternating gradient methods. 2022.
> - Y. Huang et al.. Online min-max problems with non-convexity and non-stationary. TMLR, 2023.
> - Z. Guan et al.. Online nonconvex optimization with limited instantaneous oracle feedback. COLT, 2023.
>
> Q2: No dependence on lower-level suboptimality in the regret.
>
> A2: Thank you for your constructive comments. We have the following response to this question:
>
> (1) Indeed, we have also thought about the reviewer’s suggestion for the regret definition at the beginning of this study. However, it turns out that the physical meaning of $f_t(x_t, y_t)$ or $f_t(x_t, y_{t+1})$ is not clear in bilevel optimization, whereas $f_t(x_t, y_t^*(x_t))$ is the true objective function in the outer level. And a small value of $f_t(x_t, y_t)$ or $f_t(x_t, y_{t+1})$ does not imply a small value of $f_t(x_t, y_t^*(x_t))$; namely, the bilevel problem requires to make $f_t(x_t, y)$ as small as possible particularly by $y_t^*(x_t)$ that minimizes the inner problem, not by any other values $y_t$ or $y_{t+1}$. Thus, minimizing the regret in terms of $f_t(x_t, y_t)$ or $f_t(x_t, y_{t+1})$ does not necessarily imply that $y_t$ or $y_{t+1}$ are near optimal with respect to inner loop problem, which then may not be desirable decision variables in the bilevel optimization problem at each step.
>
> (2) In fact, the sub-optimality in the lower-level decision variable has contributed to the regret through the update of $x_t$ in Equation (6), which highly depends on the quality of $y_{t+1}$. Intuitively, when $y_{t+1}$ is a more accurate estimate of $y_t^*(x_t)$, the hypergradient estimation is more accurate and the outer-level $x_{t+1}$ will be a better decision, leading to a smaller regret.
>
> Q3: Does SOBOW need for a smaller upper-level learning rate?
>
> A3: We do need a smaller learning rate for the upper-level problem. As shown in Theorem 5.7, the upper-level learning rate $\beta$ is in the order of $o(\alpha^2)$, where $\alpha$ is the lower-level learning rate.
>
> Q4: What is the motivation to not just solve the least-squares problem to sufficient optimality at each step and remove the error term in Lemma 5.5, and simplify the analysis?
>
> A4: In practice, solving the problem for $v_t^*$ to sufficient optimality can introduce high computational cost at each step, which can significantly slow down the learning process. In contrast, the computational cost can be substantially reduced without requiring high estimation accuracy of $v_t^*$ at each step in our work. Further, note that in (K. Ji et al., 2022) it is shown that the value of $Q_t$ for solving the problem for $v_t^*$ indeed has a relatively weak impact on the overall performance. For example, in (K. Ji et al., 2022), the cases between $Q=1$ and $Q=20$ have very similar performance and also similar running time. This implies that the computational cost introduced by the increasing $Q_t$ is actually negligible. Therefore, in this work we consider the sub-optimality of $v_t$ and take the estimation error for $v_t^*$ into consideration.
>
> - K. Ji, et al.. Will bilevel optimizers benefit from loops? NeurIPS, 2022.
>
> We thank the reviewer again for your insightful comments. Again, if our response resolves your concern, we will appreciate it very much if you could consider increasing the score. We will also be very happy to answer any further questions you may have.

---

> > ### Author Response · Authors · 2023-08-18
> >
> > Dear Reviewer ouS8,
> >
> > Since the author-reviewer discussion period has started for one week, and will end very soon. Could you please check our response at your earliest convenience? This way, if you have further questions, we will still have time to respond before the discussion period ends. We thank the reviewer very much in advance for your time and efforts.

---

> ### Comment · Area_Chair_bzuk · 2023-08-18
>
> Dear Reviewer ouS8: can you read the authors' response, and see if your comments are addressed?

---

> > ### Author Response · Authors · 2023-08-19
> >
> > Dear Reviewer ouS8,
> >
> > We would like to bring to your attention that your individual ratings on Soundness (3 good), Presentation (4 excellent) and Contribution (3 good) are not consistent with your final rating 3 of our paper. Your review seems to also suggest that you highly favor this paper, as reflected by your comments such as “So it is a significant contribution to have a single loop algorithm that has sublinear regret”; “The computational gains are very significant”; “This is an impressive result, making the solution even more practically useful”, etc. Your comments on the weakness part seem to have only clarification questions, which we believe we have provided convincing answers in our response.
> >
> > Could you please reconsider your final rating of the paper to make it aligned with your review and our response to your concerns. Of course, if you have any further questions, we would be very happy to address them.
> >
> > Thank you very much for your time and efforts!

---

> > > ### Comment · Reviewer_ouS8 · 2023-08-21
> > > **Main concerns addressed**
> > >
> > > I thank the authors for their detailed response to my comments and questions. My main concerns have been addressed as it seems that the choice of the regret is standard in literature. I will update my score to support this paper.

---

> > > > ### Author Response · Authors · 2023-08-21
> > > >
> > > > Many thanks for raising the score! We genuinely appreciate your constructive comments and time.

---

### Official Review · Reviewer_M56B · 2023-07-06

**Soundness:** 3 good
**Presentation:** 3 good
**Contribution:** 3 good
**Rating:** 6
**Confidence:** 3

**Summary:**

The authors consider bilevel optimization in the online setting. In this setting, we have access at iteration $t$ to the outer function $f_t$ which is assumed to be differentiable and possibly nonconvex. We also have access to the inner function $g_t$ which is assumed to be twice differentiable and strongly-convex with respect to the inner variable $y$. They propose SOBOW, an algorithm that implement approximate implicit differentiation in a single loop fashion. In SOBOW, at iteration $t$, the inner variable is updated by a gradient step and the solution of the linear system involved in the expression of the hypergradient is approximated by Conjugate Gradient steps. Then, the obtained approximate hypergradient is stored and the outer variable $x$ is updated following the opposite direction given by an average of the $K-1$ last approximate hypergradient computed. The outer variable is then projected on the constrains set $\mathcal{X}$.
The authors show that SOBOW achieves a sublinear local regret.
SOBOW is numerically compared with OGD and OAGD on an online hyper-representation learning task using a simulated dataset, and on an online hyperparameter optimization problem using the 20newsgroups dataset.

**Strengths:**

* The paper is clearly written
* The authors study online bilevel optimization which has been very little studied in the literature.
* The proposed method is theoretically grounded
* The method improves upon previous work by avoiding the evaluation of the previous functions at the current iterates.
* Numerical validation is provided on several tasks.

**Weaknesses:**

* The idea of single-loop updates was already exploited in offline context [1, 2, 3]. The authors should mention it.

* Since the authors consider a projection onto $\mathcal{X}$, this set has to be assumed closed.

* Under a closedness assumption of $\mathcal{X}$ and $\mathcal{X}$ being assumed to be bounded, the boundedness of $\nabla f$ is automatic making assumption 5.4 unecessary.

* In terms of notations, the notation $\nabla f_t(x, y^*(x))$ is confusing because it can be thought as the gradient of the function $f_t$ evaluated at the point $(x, y^*(x))$ or as the gradient of the function $x\mapsto f_t(x,y^*(x))$. Maybe it should be clearer to give a name to the function $x\mapsto f_t(x,y^*(x))$.

[1] M. Hong, H.-T. Wai, Z. Wang, and Z. Yang. A Two-Timescale Framework for Bilevel Optimization: Complexity Analysis and Application to Actor-Critic.  arXiv:2007.05170, 2021

[2] M. Dagréou, P. Ablin, S. Vaiter, and T. Moreau. A framework for bilevel optimization that enables stochastic and global variance reduction algorithms. NeurIPS, 2022.

[3] J. Li, B. Gu, and H. Huang. A Fully Single Loop Algorithm for Bilevel Optimization without Hessian Inverse. AAAI, 2022

**Questions:**

NA

---

> ### Author Rebuttal · Authors · 2023-08-09
>
> Thank you for your thorough reviews and constructive comments. We provide our response to your comments below.
>
> Q1: The idea of single-loop updates was already exploited in offline context [1, 2, 3]. The authors should mention it.
>
> A1: Thank you for bringing up these studies. We will add them in the related work in the revision.
>
> Q2: Since the authors consider a projection onto $\mathcal{X}$, this set has to be assumed closed.
>
> A2: Thank you for pointing out this missing statement. We will change the statement to “the closed convex set $\mathcal{X}$” in Assumption 5.3.
>
> Q3: Under a closedness assumption of $\mathcal{X}$ and $\mathcal{X}$ being assumed to be bounded, the boundedness of $\nabla f$ is automatic making assumption 5.4 unnecessary.
>
> A3: Many thanks for the good suggestion. We will change this assumption to a statement.
>
> Q4: In terms of notations, the notation $\nabla f_t(x, y^*(x))$ is confusing because it can be thought as the gradient of the function $f_t$ evaluated at the point $(x, y^*(x))$ or as the gradient of the function $x \rightarrow f_t(x, y^*(x))$. Maybe it should be clearer to give a name to the function $x\rightarrow f_t(x, y^*(x))$.
>
> A4: Thank you for the constructive suggestion. We will define $\Phi_t(x): x\rightarrow f_t(x, y_t^*(x))$ in the revision.

---

> > ### Comment · Reviewer_M56B · 2023-08-13
> >
> > Dear authors,
> >
> > Thank you for your answer and corrections.

---

### Official Review · Reviewer_98z6 · 2023-07-07

**Soundness:** 2 fair
**Presentation:** 3 good
**Contribution:** 2 fair
**Rating:** 5
**Confidence:** 4

**Summary:**

This paper proposed a new method for solving online bilevel problem that only required one-step $y$ update and leveraged the historical information to smooth the update. Extensive experiments are provided to validate their theories.

**Strengths:**

1.	This work is the second one considering the online bilevel optimization problem and this problem can motivate many applications.
2.	The algorithm does not require multiple inner updates and the evaluation of the current models on previous functions, so that it is more applicable to online setting.

**Weaknesses:**

1. One of the challenges unique to online bilevel optimization, as compared to its offline counterpart in this paper, lies in controlling the hypergradient estimation error, which is dependent on $ ||y_t^*(x_t)-y_{t+1}^*(x_{t+1})||^2$. Unlike in the offline case, it cannot be simply bounded by $|| x_{t+1} -x_t ||$ due to the time-variant nature of $g_t$. It is impossible to control this term without making variation assumption on the lower-level objective, so it is intriguing to see how to regulate this term by bounded variational assumption of $g_t$. However, Theorem 5.7 seems to circumvent this challenge by directly converting the hypergradient estimation error term to $V_T$ and $H_T$ — terms over which we cannot directly control. Is it possible to characterize these two terms explicitly by the variation of $g_t$? This would offer more insights into the effect on the overall online bilevel optimization.

2. The state-of-the-art work on offline bilevel optimization also adopts the three-level optimization and treats $v$ as a solution to a quadratic problem and thus can eliminate the conjugate gradient loop. As one of the contributions of this work is to reduce the multiple-step lower-level updates to one-step, it is also intriguing to see whether the conjugate gradient loop can be reduced since it is also time-consuming.

**Questions:**

Could the bounded function value in Assumption 5.3 be relaxed to merely on the feasible set? In this way, it can be derived from the bounded feasible set and Lipschitz continuity assumptions. Otherwise, bounded function value on the whole space is relatively restricted. Also, does the objective in the experiment part satisfy this assumption?

**Limitations:**

Yes

---

> ### Author Rebuttal · Authors · 2023-08-09
>
> Thank you for your thorough reviews and constructive comments. We provide our response to your comments below. If our response resolves your concern, we would greatly appreciate it if you could consider increasing your score.
>
> Q1: Is it possible to characterize these two terms explicitly by the variation of $g_t$? This would offer more insights into the effect on the overall online bilevel optimization.
>
> A1: Thank you for this insightful comment. We have the following response to this question:
>
> (1) Indeed as suggested by the reviewer, it is possible to explicitly analyze the regret in terms of the variation of $g_t$. For example, in terms of $H_T$, based on the strong convexity of the function $g_t$, we can further bound $||y_t^*(x_t)-y_{t+1}^*(x_t)||^2$ from above based on the function variation $\sup_y |g_t(x_t, y)-g_{t+1}(x_t, y)|$; in terms of $V_T$, based on the Lipschitz continuity of function $f$, we can upper bound $f_{t+1}(x, y_{t+1}^*) - f_t(x, y_t^*(x))$ based on $||y_t^*(x)-y_{t+1}^*(x)||^2$ and the function variation of $f$, i.e., $\sup_y [f_{t+1}(x, y)-f_t(x,y)]$, where the first term can be further bounded above by the function variation of $g$. We will discuss this and have a more detailed investigation in the revision.
>
> (2) In online bilevel optimization, the variation of $y_t^*(x)$ is more important since it directly affects the outer-level objective functions. Further, for strongly convex inner-level functions $g_t$, when the variation of $g_t$ is small, i.e., $|g_t(x, y)-g_{t+1}(x, y)|$ is small, the gap between $y_t^*(x)$ and $y_{t+1}^*(x)$ will not be large; when the function value $g_t$ changes significantly, as long as the variation of $y_t^*(x)$ is small, our algorithm can still guarantee a small regret. In this sense, the condition on the variation of $y_t^*(x)$ is weaker compared to the condition on the variation of $g_t$ in order to achieve a small regret.
>
> (3) Using path-length regularization to capture the variation of optimal decision variables is very common in the literature of dynamic online learning, e.g.,
>
> - M. Zinkevich. Online convex programming and generalized infinitesimal gradient ascent. ICML, 2003.
> - A. Jadbabaie et al.. Online optimization: competing with dynamic comparators. AISTATS, 2015.
> - A. Mokhtari et al.. Online optimization in dynamic environments: improved regret rates for strongly convex problems. CDC, 2016
> - T. Yang et al.. Tracking slowly moving clairvoyant: optimal dynamic regret of online learning with true and noisy gradient. ICML, 2016.
> - L. Zhang et al.. Improved dynamic regret for non-degenerate functions. NeurIPS, 2017.
> - P. Zhao et al.. Dynamic regret of convex and smooth function. NeurIPS, 2020.
>
> Note that because this variation of optimal decision variables is not controllable, we do not use this term in the design of the algorithm. Rather, the variation term is only used in the theoretical analysis to understand which factors in the system lead to a tighter bound on the regret.
>
> Q2: It is also intriguing to see whether the conjugate gradient loop can be reduced since it is also time-consuming.
>
> A2: We are not very sure about the question the reviewer asked.
>
> (1) If the reviewer refers to directly using the closed form solution of $v^*$, this solution involves the calculation of Hessian inverse which is computationally expensive.
>
> (2) If the reviewer refers to using only one step of conjugate gradient to estimate $v^*$, this is doable for offline bilevel optimization with time-invariant objective functions, where slow changes of the variables can still make progress to solve the optimization problem. However, in online bilevel optimization, since the estimation error of $v_t^*$, i.e., $||v_t^Q-v_t^*||$,  depends on the estimation error of $v^*_{t-1}$ in the last round and the variation of $v_t^*$, i.e., $||v_{t-1}^*-v_t^*||$, we need to make sure that  $||v_{t-1}^*-v_t^*||$ will decay with $t$ in order to achieve a sublinear regret with one-step conjugate gradient. In the offline case, $||v_{t-1}^*-v_t^*||$ only depends on $||x_{t-1}-x_t||$ which can decay gradually; however, in the online case, $||v_{t-1}^*-v_t^*||$ also depends on the function variations, such that additional conditions may be needed for achieving a sublinear regret. To summarize, reducing the number of conjugate gradient steps is an interesting open problem, which is beyond the scope of this paper, but one that we plan to investigate for future work.
>
> Q3: Could the bounded function value in Assumption 5.3 be relaxed to merely on the feasible set? Also, does the objective in the experiment part satisfy this assumption?
>
> A3: (1) Yes, Assumption 5.3 can be relaxed that the function value on $(x, y^*(x))$, i.e., $f_t(x, y^*(x))$, is bounded, which holds given the bounded feasible set, Lipschitz continuity and also the bounded set of $y_t^*(x)$. Here $y^*(x)$ is generally assumed to be bounded in bilevel optimization such that the lower-level problem can be solved to a certain accuracy. (2) Yes, the objective in the experiments satisfies this assumption. For example, in the online hyper-representation learning problem, since both the value of data samples and the feasible set of the decision variables are bounded, the function values are bounded.
>
> We thank the reviewer again for your insightful comments. Again, if our response resolves your concern, we will appreciate it very much if you could consider increasing the score. We will also be very happy to answer any further questions you may have.

---

> > ### Comment · Reviewer_98z6 · 2023-08-17
> >
> > Thank you for your clarification; it has addressed some of my questions. I think that controlling the term $\\|y_t^*\left(x_t\right)-y_{t+1}^*\left(x_t\right)\\|^2$ by bounded variational assumption of $g_t(x,y)$ remains crucial to online bilevel optimization. As you also concur with its potential solution, it might be better to incorporate a more detailed discussion and a rigorous theory on this topic in the current version.
> >
> > Regarding Q2, I'm referencing the fully single-loop techniques [1]-[3] in offline bilevel optimization, where $v$ is treated as another optimization variable, akin to $y$. Given your strategy to reduce the number of loops for optimizing $y$, could a similar approach be applied to $v$?
> >
> > [1] A Fully Single Loop Algorithm for Bilevel Optimization without Hessian Inverse. AAAI 2022.
> >
> > [2] A framework for bilevel optimization that enables stochastic and global variance reduction algorithms. NeurIPS 2022.
> >
> > [3] Amortized Implicit Differentiation for Stochastic Bilevel Optimization. ICLR 2022.

---

> > > ### Author Response · Authors · 2023-08-18
> > > **Response to the further comment**
> > >
> > > Q1: It might be better to incorporate a more detailed discussion and a rigorous theory on this topic in the current version.
> > >
> > > A1: Thank you for the advice. We have developed the following theorem by using the function variations:
> > >
> > > **Theorem. Suppose that Assumptions 5.1-5.4 hold. Let $V_g=\sum_{t=1}^T \sup |g_{t+1}(x, y)-g_{t}(x, y)|$ and $V_f=\sum_{t=1}^T \sup [f_{t+1}(x, y)-f_t(x, y)]$. Under the same conditions on $\lambda$, $\alpha$, $Q_t$, $\eta$ and $\beta$ with Theorem 5.7, we can have $BLR_w(T)\leq O\left(\frac{T}{\beta W} + \frac{V_f}{\beta} + V_g + \frac{\sqrt{T V_g}}{\beta} \right)$.**
> > >
> > > A more detailed analysis is as follows:
> > >
> > > (1) For $H_{2,T}$, based on the strong convexity of $g_t$, we can show that $||y_{t+1}^*(x)-y_{t}^*(x)||^2 \leq \frac{2}{\mu_g} \sup |g_{t+1}(x, y)-g_{t}(x, y)|$;
> > >
> > > (2) For $V_{1,T}$, we can show that $f_{t+1}(x, y^*_{t+1}(x))-f_t(x,y_t^*(x))\leq L_0 ||y^*_{t+1}(x)-y_t^*(x)||+\sup [f_{t+1}(x, y)-f_t(x, y)]$, such that Line 667 (Lemma G.3) in Appendix can be upper bounded by $\frac{2MT}{W}+L_0\sqrt{\frac{2T}{\mu_g}}\sqrt{\sum_{t=1}^T \sup |g_{t+1}(x, y)-g_{t}(x, y)|}+ \sum_{t=1}^T \sup [f_{t+1}(x, y)-f_t(x, y)]$;
> > >
> > > (3) Based on these, if we denote $V_g=\sum_{t=1}^T \sup |g_{t+1}(x, y)-g_{t}(x, y)|$ and $V_f=\sum_{t=1}^T \sup [f_{t+1}(x, y)-f_t(x, y)]$ to capture the function variations, we can have the overall regret as $O\left(\frac{T}{\beta W} + \frac{V_f}{\beta} + V_g + \frac{\sqrt{T V_g}}{\beta} \right)$. In this case, a sublinear regret will be achieved if both $V_g$ and $V_f$ are $o(T)$ for suitably selected $W$. As mentioned in our previous response, the condition on the variation of $y_t^*(x)$ is weaker compared to the condition on the variation of $g_t$ in order to achieve a small regret. For example, suppose $W= \omega(T)$ and the function variation of $f_t$ is very small, to achieve a regret of $O(T^{3/4})$,  $H_{2,T}=O(T^{3/4})$ is sufficient, while we need a stricter condition on the variation of $g_t$, i.e., $V_g=O(T^{1/2})$.
> > >
> > > We will add the theorem and more detailed discussions on this in the final version per the reviewer’s suggestion. Once again, we do not use the terms $H_{2,T}$ and $V_{1,T}$ in the algorithm. Rather, the variation term is only used in the theoretical analysis to understand which factors in the system lead to a tighter bound on the regret.
> > >
> > > Q2: Given your strategy to reduce the number of loops for optimizing $y$, could a similar approach be applied to $v$?
> > >
> > > A2: This does not appear to be the case because the objective functions in the offline setting such as in [1]-[3] are **time-invariant**, while the objective functions in the online setting (such as ours) are **time-varying**. Hence, in the offline setting, it is easier to control the error even with only one step conjugate gradient estimate of $v^*$ because of the offline time-invariant setup. In contrast, because of the time-varying nature of the objective function in the online setting, controlling the error with only one step conjugate gradient becomes extremely difficult. More details are provided below to explain this difficulty.
> > >
> > > (1) In the current work, we seek to reduce the number of steps for updating $y_t$ so that our algorithm can also work under limited knowledge of the function $g_t$. But this can result in a large estimation error for the hypergradient at each step. To control this error, we carefully control the estimation errors of $y_t^*$ and $v_t^*$ together, such that the summation of $||y_t^*(x_t)-y_{t+1}||^2$ and $||v_t^*-v_t^Q||^2$ will decay in order to achieve a sublinear regret under function variations. We achieve this by increasing the estimation accuracy for $v_t$ (note that this does not require more information about the function $g_t$), which compensates the large estimation error of $y_t^*$ due to a single step update.
> > >
> > > (2) When further reducing the number of update steps for $v_t$, we still need to jointly control the estimation error of $y_t^*$ and $v_t^*$. But the strategy we take above doesn’t work anymore since the estimation error of $v_t^*$ is also large. Besides, the warm start strategy used in offline bilevel optimization will not work due to the time-varying functions in online bilevel optimization. In particular, the estimation error of $v_t^*$ depends on both the update of $x_t$, $y_t^*$ and the function variations including both outer-level function $f_t$ and inner-level function $g_t$. This will largely complicate the analysis and make achieving a sublinear regret highly nontrivial. Investigating this problem is very interesting but is worth to be considered as an independent future work.
> > >
> > > Finally, if our response resolves your concerns to a satisfactory level, we kindly ask the reviewer to reconsider raising the score of your evaluation. Certainly, we are more than happy to address any further questions that you may have during the discussion period. We thank the reviewer again for the helpful comments and suggestions for our work.

---

> > > > ### Comment · Reviewer_98z6 · 2023-08-19
> > > >
> > > > Thank you for your careful response and I appreciate your effort. Given that my questions are solved, I'm willing to raise my score to 5.

---

> > > > > ### Author Response · Authors · 2023-08-19
> > > > >
> > > > > Thank you so much for all the valuable comments and raising the score!

---

### Official Review · Reviewer_Ue3m · 2023-07-07

**Soundness:** 3 good
**Presentation:** 3 good
**Contribution:** 4 excellent
**Rating:** 7
**Confidence:** 3

**Summary:**

This work studies the online bilevel optimization problem with nonstationary and time-varying objective functions. This line of research can cover applications with online nature like online meta-learning, online hyperparameter tuning, wireless networks. Compared to widely studied offline bilevel problem, the studied one has challenges like limited information, hypergradient computation, changing objectives. The authors propose a single-loop online bilevel optimizers called SOBOW based on online nonconvex optimization and window averaging, and further show it can attain a sublinear regret. Some experiments are provided to justify the effectiveness of the proposed method.

**Strengths:**

1. Bilevel optimization has been studied intensively mainly in the offline setting where all objective functions are fixed and known. It has been much less explored in the online and nonstationary setting. This work seems to provide a simple and good solution.

2. The authors have done a good job in discussing the underlying challenges and the drawbacks of existing method in [54]. How to design a good online bilevel optimizer with limited queries, efficient hypergradient computation, guaranteed regret turns out to be nontrivial.

3. Technically, this work needs to cope with 1) the intersection among three variables $x,y,v$ in online manner,  2) time-smoothed gradient updates, 3) biased gradient estimation, which may be straightforward.

**Weaknesses:**

1. The hypergradient estimation contains second-order derivatives. Will them cost a lot in practical online settings? Is it possible to design fully first-order method without matrix-vector computations? There are some recent progress towards the Hessian-free bilevel optimization.

2. The authors do not compare the rate of their method with OAGD.

**Questions:**

See weaknesses.

**Limitations:**

Yes

---

> ### Author Rebuttal · Authors · 2023-08-09
>
> Thank you for your thorough reviews and constructive comments. We provide our responses to your comments below.
>
> Q1: Is it possible to design fully first-order methods without matrix-vector computations?
>
> A1: Many thanks for the insightful comments and suggestions. Our current algorithm involves matrix-vector computations to deal with second-order derivatives, which is fairly efficient in our experiments. As the reviewer suggested, the computational cost of the algorithm can be further reduced via fully first-order methods or Hessian-free design. We describe these ideas as follows.
>
> To the best of our knowledge, the recent first-order approaches mainly follow two strategies: (1) replace the second-order term in the hypergradient estimation with zeroth-order estimations; (2) reformulate the bilevel problem to a single-level constrained optimization problem and use first-order methods to solve the reformulated problem.
>
> For the first strategy, applying it to our approach is straightforward by replacing the second-order term in online hypergradient via their zeroth-order estimations. For the second strategy, we need to reformulate the bilevel problem at each round to a constrained optimization problem, and then develop an online algorithm for solving such a reformulated nonconvex constrained problem. The primal-dual methods can be leveraged to solve such a constrained problem. Particularly, a recently developed single-loop algorithm for nonconvex constrained problems in (S. Lu et al., 2022) can be leveraged to develop a single-loop online algorithm and analyze its regret performance.
>
> - S. Lu et al.. A single-loop gradient descent and perturbed ascent algorithm for nonconvex functional constrained optimization. ICML, 2022.
>
> Q2: The authors do not compare the rate of their method with OAGD.
>
> A2: Thanks for the question. We have the following response to this question:
>
> (1) Since our work uses a different regret definition from that in OAGD, it would not be fair for a direct comparison between the regret bounds.
>
> (2) Instead, as seen in Figure 1, we compare the performance of our algorithm and OAGD using our definition of regret. And we have also conducted experiments to compare the performance between our algorithm and OAGD using the regret defined in OAGD (Figure 2 in Appendix). In both cases, we can see that our algorithm achieves comparable regret performance with OAGD, but with a much shorter runtime.

---

> > ### Comment · Reviewer_Ue3m · 2023-08-16
> >
> > Thank you for the response. It clarifies my questions. I remain the score.

---

### Decision · Program_Chairs · 2023-09-21

**Decision:**

Accept (poster)

**Comment:**

This paper studies online bilevel optimization, which is an important topic yet still less developed. The paper proposes a single-loop algorithm for solving this problem and provides analysis for the regret. The method improves existing method from several aspects. It is a nice contribution to the field.